# Reshaping Reasoning in LLMs: A Theoretical Analysis of RL Training Dynamics through Pattern Selection

**Xingwu Chen**[*]
School of Computing & Data Science
The University of Hong Kong
`xingwu@connect.hku.hk`

**Tianle Li**[*]
Institute of Data Science
The University of Hong Kong
`tianleli@connect.hku.hk`

**Difan Zou**
School of Computing & Data Science
Institute of Data Science
The University of Hong Kong
`dzou@cs.hku.hk`

## Abstract

While reinforcement learning (RL) demonstrated remarkable success in enhancing the reasoning capabilities of language models, the training dynamics of RL in LLMs remain unclear. In this work, we provide an explanation of the RL training process through empirical analysis and rigorous theoretical modeling. First, through systematic reasoning-pattern-level and token-level analysis across the RL training process, we show that while different reasoning patterns exhibit relatively stable success rates during training, RL primarily optimizes a sparse subset of critical tokens, thereby reshaping reasoning pattern distributions to affect model performance. Building on these empirical insights, we develop a theoretical framework to understand the training dynamics of RL with two typical rewards: verifiable reward (RLVR) and model's internal feedback (RLIF). For RLVR, we analyze the training dynamics under two special cases: one where models readily converge to optimal reasoning strategies, and another where optimization becomes challenging, revealing that the base model's reasoning quality is crucial for determining convergence behavior. For RLIF, we examine how internal rewards initially improve model performance but can potentially lead to degradation with continued training. Extensive experiments validate our findings, advancing both theoretical understanding and practical applications of RL in language model enhancement.

## 1 Introduction

Recently, state-of-the-art reasoning models such as Gemini2.5 (Comanici et al., 2025), Qwen3 (Yang et al., 2025), and DeepSeek-R1 (Guo et al., 2025) demonstrate exceptional performance on complex logical tasks including mathematics (Shao et al., 2024; Zeng et al., 2025; Yu et al., 2025) and programming (Zhu et al., 2024; Yang et al., 2025; Comanici et al., 2025). Reinforcement learning (RL) serves as a key technique behind this success, demonstrating the potential to elevate model capabilities to a new level.

The success of RL has triggered research into its underlying mechanisms for LLMs. By comparing pass@k performance, Yue et al. (2025a) show that models post-trained with RL struggle to surpass base models, suggesting that RL may not elicit fundamentally new reasoning patterns. From an entropy perspective, Cui et al. (2025a); Zhang et al. (2025b) theoretically prove that RL-based methods can reduce policy entropy, with Cui et al. (2025a) also empirically establishing a connection between model performance and policy entropy. Wang et al. (2025a); Huan et al. (2025); Meng et al. (2026) further demonstrate that RL primarily optimizes a sparse subset of critical tokens. Despite

---

[*]Equal contribution.

these various efforts to understand RL mechanisms, the underlying dynamics of the RL training process remain incompletely understood both empirically and theoretically.

To understand the RL training process, we first present a systematic reasoning-pattern-level and token-level analysis across RL training. Specifically, focusing on the training dynamics, we not only examine the ranking shifts across training, but also use LLM-based and rule-based methods to extract and classify reasoning patterns from models' responses, analyzing the corresponding success rates and distributions during training. Compared with previous works (Huan et al., 2025; Yue et al., 2025a), our experiments provide clearer and more compelling evidence demonstrating that RL primarily optimizes a sparse subset of critical tokens, thereby *reshaping reasoning pattern distributions* to affect model performance. Moreover, we find that the intrinsic success rate of individual patterns remains relatively stable. These experimental insights inspire us to develop a mathematical framework to theoretically understand the RL training process.

Based on our empirical findings, we further develop a theoretical framework that conceptualizes reasoning as a two-stage question-reason-answer process $q \rightarrow r \rightarrow a$: (1) reasoning pattern selection based on the question, i.e., $\pi(r|q)$, and (2) answer generation based on the chosen pattern, i.e., $\pi(a|r, q)$. Using this framework, we theoretically analyze the training dynamics of RL in LLMs with two typical reward types: verifiable reward (RLVR) (Guo et al., 2025; Shao et al., 2024) and the model's internal feedback (RLIF) (Zhao et al., 2025b; Agarwal et al., 2025). For RLVR, we show that it can converge to the reasoning pattern $r^*$ with the highest success rate, which precisely matches our empirical observations. Moreover, we characterize two distinct convergence regimes: models with strong initial reasoning quality demonstrate rapid convergence to optimal patterns, while weaker models face entanglement-stage optimization challenges. For RLIF, we provide an explanation of why RL with internal rewards can improve model performance, and we also show that RLIF may ultimately converge to a state with worse performance than the base model, aligning with our empirical findings. Additional experiments validate our theoretical analysis.

The main contributions of this paper are highlighted as follows:

- We conduct systematic reasoning-pattern-level and token-level analysis across the RL training process. Through examining next token prediction ranking shifts, combined with LLM-based and rule-based reasoning pattern analysis, we provide clearer and more compelling evidence for understanding RL training dynamics compared with previous works (Huan et al., 2025; Yue et al., 2025a). Our experiments demonstrate that RL primarily optimizes a sparse subset of critical tokens, thereby *reshaping reasoning pattern distributions* to affect model performance, while the intrinsic success rate of individual patterns remains relatively stable during training.

- We develop a formal two-stage mathematical framework that models reasoning as $q \rightarrow r \rightarrow a$ (question-reason-answer) and theoretically analyze training dynamics for two typical RL-based approaches. For RLVR, we prove convergence to the reasoning pattern with the highest success rate and characterize two distinct regimes: rapid convergence for strong base models versus optimization challenges during entanglement stages for weaker models. For RLIF, we provide theoretical justification for the performance improvements at early training stages, and explain why such methods may ultimately converge to worse performance than the base model.

- We validate our theoretical analysis through additional case studies. Our framework provides practical insights for understanding and improving RL-based LLM post-training, bridging the gap between empirical observations and theoretical understanding of RL training dynamics in LLMs.

## 2 RELATED WORKS

**Reinforcement Learning for LLMs** Reinforcement learning has demonstrated remarkable success in enhancing large language models (LLMs), particularly in aligning models with human preferences (Ouyang et al., 2022; Zhu et al., 2023) and in solving complex mathematical and programming tasks (Shao et al., 2024; Jaech et al., 2024; Lambert et al., 2024). A central component of RL is the reward model, which traditionally relies on human-annotated datasets (Rafailov et al., 2023; Ouyang et al., 2022; Achiam et al., 2023) with extensive training. Recently, the paradigm has shifted toward leveraging more easily obtainable and verifiable rewards, such as in reinforcement learning with verifiable rewards (RLVR) (Shao et al., 2024; Yang et al., 2025; Guo et al., 2025) and reinforcement learning with internal feedback (RLIF) (Zhao et al., 2025b; Agarwal et al., 2025). Moreover, the

success of RL has spurred the development of new RL algorithms, including GRPO (Shao et al., 2024), DAPO (Yu et al., 2025), DERL(Cheng et al., 2025), and VAPO (Yue et al., 2025b).

**Theoretical Analyses and Mechanism Interpretation of RL for LLMs.** While existing theoretical analyses have provided valuable insights into the general mechanics of large language models (Azar et al., 2024; Aminian et al., 2025; Chen and Zou, 2024; Chen et al., 2024; Li et al., 2025; Zhang et al., 2026; 2025a), the remarkable success of RL in enhancing LLMs has triggered a dedicated wave of research focused specifically on its theoretical foundations and mechanism interpretation. Initial efforts in this area mainly concentrated on high-level RL dynamics, such as reward design Scheid et al. (2024); Huang et al. (2025); Xu et al. (2024) and the development of novel training algorithms Xiong et al. (2024); Das et al. (2024); Ji et al. (2024). As RL paradigms shift towards simpler reward structures like RLVR and RLIF, researchers have sought to understand the fundamental reasons behind its effectiveness from various perspectives. Shao et al. (2025) revealed that spurious rewards enhance reasoning by unlocking latent skills from pretraining, while Agarwal et al. (2025) explained performance improvements through entropy minimization. Gandhi et al. (2025) identified specific cognitive behaviors, such as backtracking, that contribute to improved reasoning capabilities. Further insights came fromZhao et al. (2025a), who demonstrated that RL fine-tuning amplifies pre-trained behaviors, leading to convergence towards dominant output formats, a phenomenon further analyzed by Wang et al. (2025b); Cui et al. (2025b) through the lens of entropy collapse. Unlike previous work, our study presents an analysis of RL training dynamics from the perspective of reasoning pattern selection, supported by a mathematical framework with theoretical analysis.

## 3 PRELIMINARIES

**Reinforcement Learning for LLMs** Let $\pi_{\boldsymbol{\theta}}$ be a language model with parameters $\boldsymbol{\theta}$, which serves as the policy to be optimized. Given an input question $\boldsymbol{x} = (x_0, x_1, \ldots, x_n)$, the policy $\pi_{\boldsymbol{\theta}}$ generates an answer $\boldsymbol{y} = (y_0, y_1, \ldots, y_m)$. The optimization objective can be formulated as:

$$\phi_{\mathrm{RL}}(\boldsymbol{\theta}) = \mathbb{E}_{\boldsymbol{x} \sim \mathcal{D}, \boldsymbol{y} \sim \pi_{\boldsymbol{\theta}}(\cdot | \boldsymbol{x})} \left[ r_{\phi}(\boldsymbol{x}, \boldsymbol{y}) \right] - \beta \mathbb{D}_{\mathrm{KL}} \left[ \pi_{\boldsymbol{\theta}}(\boldsymbol{y} \mid \boldsymbol{x}) \, \| \, \pi_{\mathrm{ref}}(\boldsymbol{y} \mid \boldsymbol{x}) \right], \quad (3.1)$$

where $\pi_{\mathrm{ref}}$ is the base reference policy, and $\beta$ is a hyperparameter that controls the KL divergence to prevent excessive deviation between $\pi_{\boldsymbol{\theta}}$ and $\pi_{\mathrm{ref}}$.

The reward function $r_{\phi}(\boldsymbol{x}, \boldsymbol{y})$ can be implemented in various formats, such as a trained reward model Ouyang et al. (2022) or a rule-based scoring function Shao et al. (2024). In this paper, we focus on verifiable reward (RLVR) (Guo et al., 2025; Shao et al., 2024) and the model's internal feedback (RLIF) (Zhao et al., 2025b; Agarwal et al., 2025).

In RLVR, the reward function $r_{\phi}(\boldsymbol{x}, \boldsymbol{y})$ directly evaluates whether the answer $\boldsymbol{y}$ matches the correct answer to question $\boldsymbol{x}$. A typical reward function is defined as:

$$r_{\phi}(\boldsymbol{x}, \boldsymbol{y}) = \begin{cases} 1 & \text{if } \boldsymbol{y} = \text{the ground truth of } \boldsymbol{x}, \\ 0 & \text{otherwise.} \end{cases} \quad (3.2)$$

Unlike RLVR, RLIF (Zhao et al., 2025b; Agarwal et al., 2025) leverages rewards based solely on intrinsic model-derived signals. Following Zhao et al. (2025b), we consider the RLIF reward as the negative average KL divergence between a uniform distribution $U$ over the vocabulary $\mathcal{V}$ and the model's next-token distribution

$$r_{\phi}(\boldsymbol{x}, \boldsymbol{y}) = \frac{1}{|\boldsymbol{y}|} \sum_{i=1}^{|\boldsymbol{y}|} \mathbb{D}_{\mathrm{KL}}(U || \pi_{\theta}(\cdot | \boldsymbol{x}, \boldsymbol{y}_{<i})) = -\frac{1}{|\boldsymbol{y}| \cdot |\mathcal{V}|} \sum_{i=1}^{|\boldsymbol{y}|} \sum_{j=1}^{|\mathcal{V}|} \log(|\mathcal{V}| \cdot \pi_{\theta}(\cdot | \boldsymbol{x}, \boldsymbol{y}_{<i})). \quad (3.3)$$

In this paper, we aim to understand the training dynamics of both RLIF and RLVR through empirical and theoretical analysis.

## 4 EXPERIMENTAL EXPLORATION FOR RL TRAINING

To understand the RL training process, we conduct systematic experiments and analysis across RL training with different rewards. Our experiments begin with a high-level overview of the training procedure, revealing that RLVR yields steady improvements, whereas RLIF can be unstable and even

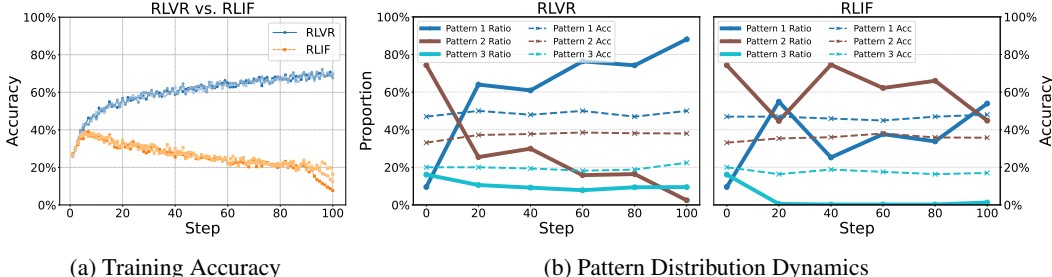

(a) Training Accuracy

(b) Pattern Distribution Dynamics

Figure 1: *(a)* The training procedures using RLVR and RLIF, showing the performance on the MATH dataset. A key finding is that RLVR provides stable gains, while RLIF causes model performance to initially increase and then decrease. We conduct multiple rounds of experiments for each training paradigm by setting the random seed. *(b)* The reasoning-pattern level comparison. The solid line represents the proportion of a certain pattern in the responses among all patterns (left vertical axis). The dotted line represents the accuracy corresponding to the pattern (right vertical axis). During RLVR, the model gradually adopts patterns with higher accuracy and reduces the use of patterns with lower accuracy, while RLIF exhibits unstable training dynamics. For ease of observation, we sort and name the different patterns from high to low according to their corresponding accuracy.

degrade performance. To dissect these dynamics, we perform a reasoning-pattern level analysis, which shows that RLVR's success stems from its tendency to adopt reasoning patterns with higher success rates, while RLIF fails to specialize in these patterns. Finally, a token-level analysis investigates the underlying mechanism for these reasoning pattern dynamics, revealing that they are driven by changes in the probability ranks of a surprisingly small fraction of tokens. Compared with previous works (Huan et al., 2025; Yue et al., 2025a), our experiments provide clearer and more compelling evidence of these RL training dynamics, demonstrating that RL primarily optimizes a sparse subset of critical tokens, thereby *reshaping reasoning pattern distributions* to affect model performance.

## 4.1 TRAINING PROCEDURE OVERVIEW FOR RLVR AND RLIF

We first analyze the complete training procedures for both RLVR and RLIF. For a controlled comparison, we select Qwen2.5-3B (Yang et al., 2024) as the base model and train it on the MATH dataset (Hendrycks et al., 2021), keeping all other settings identical for both algorithms.

As illustrated in Figure 1a, the two methods exhibit markedly different performance trajectories. RLVR leads to a stable training process, where model performance continuously improves and gradually converges throughout training. In contrast, while RLIF initially improves performance in the early stages, continued training could lead to a performance drop, sometimes resulting in a model that is worse than the original base model. These results motivate us to conduct a more systematic, fine-grained empirical analysis to understand the underlying RL training dynamics.

## 4.2 REASONING-PATTERN LEVEL ANALYSIS

To understand the performance disparities observed between RLVR and RLIF, we conduct a fine-grained analysis at the reasoning-pattern level. To define and extract these patterns, we first collect responses from the base model and employ GPT-4o (Hurst et al., 2024) to group them into distinct categories based on keywords and logical structure, also generating a description for each pattern. We then use these classifications to analyze the reasoning pattern distribution and their corresponding success rates throughout the RL training procedure.

Specifically, we analyze the dynamics of reasoning patterns and their accuracy during both RLVR and RLIF training. Figure 1b illustrates our analysis on responses from Number Theory tasks from the MATH dataset (Hendrycks et al., 2021), with additional results for varying models and datasets provided in Appendix B. Our analysis reveals three key findings:

- RLVR-trained models consistently shift towards adopting reasoning patterns with higher success rates, explaining why the model's overall accuracy steadily improves with RLVR.

- The reasoning pattern distribution for RLIF exhibits unstable training dynamics and fails to specialize in more effective patterns.

- The success rate of any individual reasoning pattern remains stable throughout the training process for both methods.

This analysis indicates that the performance difference between RLVR and RLIF stems from their distinct reasoning pattern dynamics. We therefore conduct further experiments to understand the underlying mechanisms driving these changes during RL training.

## 4.3 Token-level Analysis

To understand the mechanisms behind the reasoning pattern dynamics observed during RL training, inspired by Huan et al. (2025); Yue et al. (2025a), we further conduct a token-level analysis. We first sample responses from the base model for questions from the GSM8K (Cobbe et al., 2021), MATH (Hendrycks et al., 2021), and AIME (Codeforces) datasets. Then, for each token position in these responses, we examine the corresponding token ranks in both the base and the RL-enhanced models. Here, a token's rank refers to the position of its probability among all vocabulary tokens, given the preceding context. Our analysis, summarized in Table 1, reveals that the ranks at individual token positions remain largely stable, with fewer than 10% of the positions in each response experiencing a rank shift in most cases. This finding indicates that RL selectively modifies the probabilities at a sparse set of critical decision points while leaving the majority of the reasoning process unchanged.

Our experiments provide clean and compelling evidence for RL training dynamics: *RL primarily optimizes a sparse subset of critical tokens, thereby reshaping reasoning pattern distributions to affect model performance. Moreover, the intrinsic success rate of individual patterns remains relatively stable during training.*

These empirical findings not only offer a deeper understanding of RL training dynamics but also provide insights for building a mathematical framework to conduct further theoretical analysis.

Table 1: The Ranking Change Ratio after RL.

| Task | Method | Step20 | Step40 | Step60 | Step80 | Step100 |
|------|--------|--------|--------|--------|--------|---------|
| GSM8K | RLVR | 5.2% | 6.0% | 6.6% | 7.1% | 7.3% |
| | RLIF | 6.8% | 8.0% | 8.9% | 9.5% | 10.1% |
| MATH | RLVR | 5.3% | 5.8% | 6.1% | 6.5% | 6.6% |
| | RLIF | 6.1% | 7.0% | 7.7% | 8.3% | 8.8% |
| AIME24 | RLVR | 5.1% | 5.7% | 5.9% | 6.1% | 6.3% |
| | RLIF | 5.6% | 6.5% | 7.4% | 8.2% | 8.6% |

## 5 Theoretical Characterization of RL Training Dynamics

Beyond empirical observations, we take a further step toward theoretically understanding the RL training process. In this section, we first build a theoretical framework based on experimental insights. We abstract the model's reasoning procedure as a two-step process: first selecting a reasoning pattern, then performing answer deduction, where RL specializes in optimizing the first part. Based on this framework, we provide a theoretical analysis for our observations of RL with different rewards. For RLVR, we analyze its training dynamics and examine two special cases with distinct optimization behaviors. For RLIF, we analyze why RL with internal rewards can improve model performance while also explaining why such methods may ultimately result in models with worse performance than the base model.

## 5.1 A Theoretical Framework for Reasoning Models

**Abstract Reasoning Process.** We formalize the reasoning process as follows: given a question $q$, the model (1) selects a reasoning pattern $r$ from candidate patterns $\mathcal{R} = \{r_1, r_2, \dots\}$ and (2) generates a final answer $a \in \mathcal{A} = \{a_1, a_2, \dots\}$ accordingly. The model selects reasoning pattern $r_i$ with probability $p(r_i|q)$, and each reasoning pattern has a distinct success rate. Within this framework,

we can reformulate the RLIF optimization objective (Eq. 3.3) as follows[1]:

$$r_\phi(\{\boldsymbol{r}, \boldsymbol{a}\}, q) := - \left( \frac{1}{|\mathcal{R}|} \sum_{\boldsymbol{r}_i \in \mathcal{R}} \log\left(|\mathcal{R}| \cdot \pi_\theta(\boldsymbol{r}_i|q)\right) + \frac{1}{|\mathcal{A}|} \sum_{\boldsymbol{a}_j \in \mathcal{A}} \log\left(|\mathcal{A}| \cdot \pi_\theta(\boldsymbol{a}_j|q, \boldsymbol{r})\right) \right) \quad (5.1)$$

**Policy Parameterization.** Let $\mathcal{V} = \{\boldsymbol{q}\} \cup \mathcal{R} \cup \mathcal{A}$ denote the vocabulary set and $\mathcal{Y} = (r, a)_{r \in \mathcal{R}, a \in \mathcal{A}}$ represent the set of output sequences. Given a question $\boldsymbol{q}$, the language model produces a distribution over output sequences $\boldsymbol{y} \in \mathcal{Y}$ autoregressively:

$$(\text{general policy}) \qquad \pi_{\boldsymbol{\theta}}(\boldsymbol{y}|\boldsymbol{x}) = \prod_{l=1}^{|\boldsymbol{y}|} \pi_{\boldsymbol{\theta}}(\boldsymbol{y}_l|\boldsymbol{x}, \boldsymbol{y}_{<l}) = \prod_{l=1}^{|\boldsymbol{y}|} \text{softmax}(f_{\boldsymbol{\theta}}(\boldsymbol{x}, \boldsymbol{y}_{<l}))_{\boldsymbol{y}_l}, \qquad (5.2)$$

where $f_{\boldsymbol{\theta}} : \mathcal{V} \rightarrow \mathbb{R}^{|\mathcal{V}|}$ is a function parameterized by $\boldsymbol{\theta}$, and the model predicts the $l$-th token based on the previous context $\boldsymbol{y}_{<l}$. The next token follows the distribution $\text{softmax}(\boldsymbol{z})_v := \exp(\boldsymbol{z}_v) / \sum_{v' \in \mathcal{V}} \exp(\boldsymbol{z}_{v'})$ for $\boldsymbol{z} = f_{\boldsymbol{\theta}}(\boldsymbol{x}, \boldsymbol{y}_{<l})$.

Due to the complexity of practical models, establishing optimization guarantees for understanding RL training dynamics has proven very challenging (Agarwal et al., 2021; Li et al., 2021). Following previous works (Razin et al., 2025; Mei et al., 2020; Cui et al., 2025b; Zhang et al., 2025b), we consider a *tabular policy* parameterization, which can be viewed as a special case of Eq. 5.2 where each output is assigned its own trainable logit for the corresponding last token, i.e., for $\boldsymbol{\theta} \in \mathbb{R}^{|\mathcal{V}| \times |\mathcal{V}|}$:

$$(\text{tabular policy}) \qquad \pi_{\boldsymbol{\theta}}(\boldsymbol{y}_l|\boldsymbol{x}, \boldsymbol{y}_{<l}) = \pi_{\boldsymbol{\theta}}(\boldsymbol{y}_l|\boldsymbol{y}_{l-1}) = \text{softmax}(\boldsymbol{\theta}_{:, \boldsymbol{y}_{l-1}})_{\boldsymbol{y}_l}, \qquad (5.3)$$

where $\boldsymbol{\theta}_{:, \boldsymbol{y}_{l-1}} \in \mathbb{R}^{|\mathcal{V}|}$ is the column of $\boldsymbol{\theta}$ corresponding to $\boldsymbol{y}_{l-1}$.

**Optimization for RL.** For our analysis of training dynamics, we consider the tabular policy from Eq.5.3 with the optimization objective $\phi_{\text{RL}}(\boldsymbol{\theta})$ in Eq.3.1. We analyze the policy gradient in the small learning rate limit using gradient flow:

$$\frac{d}{dt}\boldsymbol{\theta}(t) = \nabla\phi_{\text{RL}}(\boldsymbol{\theta}(t)), \quad t \geq 0 \qquad (5.4)$$

where $\boldsymbol{\theta}(t)$ represents the parameters of the policy $\pi_{\boldsymbol{\theta}(t)}$ at training time $t$, initialized with $\boldsymbol{\theta}(0) = \boldsymbol{\theta}_{\text{ref}}$.

Our experiments revealed that success rates for individual reasoning patterns often remain stable during training. We attribute this to the model's architectural constraints: optimizing the mapping from questions to reasoning patterns is substantially easier than optimizing the path from reasoning patterns to final answers. To formalize this observation, we introduce the following assumption:

**Assumption 5.1.** *The success rate for each reasoning pattern $\boldsymbol{r}_i \in \mathcal{R}$ to provide the correct answer $\boldsymbol{r}^*$, defined as $\pi_{\boldsymbol{\theta}}(\boldsymbol{a}^*|\boldsymbol{q}, \boldsymbol{r}_i)$, remains constant during training.*

In the remaining part of this section, we adopt Assumption 5.1 and use $p^*(\boldsymbol{r}) = \pi_{\boldsymbol{\theta}}(\boldsymbol{a}^*|\boldsymbol{q}, \boldsymbol{r})$ to denote the fixed success rate for the given pattern $\boldsymbol{r}$. We first derive the optimal policy for the RL objective (Eq.3.1) under the general autoregressive policy (Eq.5.2). We then analyze RLVR and RLIF training dynamics using the tabular policy (Eq. 5.3).

## 5.2 THE OPTIMAL POLICY FOR RL

Our empirical results demonstrate that RLVR improves model performance steadily through incentivizing reasoning patterns with a higher success rate, while the RLIF demonstrates an unstable improvement for the model, here, we first provide a theoretical explanation for the optimal policy for RLVR and RLIF optimizing objective:

**Proposition 5.2.** *Suppose we maximize the RL objective (Eq 3.1) using a general autoregressive policy (Eq 5.2) and Assumption 5.1 holds. Then, the optimal policy satisfies:*

$$\pi_{\text{opt}}(\boldsymbol{r}|\boldsymbol{q}) = \frac{1}{Z} \exp\left(\frac{1}{\beta} R(\boldsymbol{r})\right) \pi_{\boldsymbol{\theta}_{\text{ref}}}(\boldsymbol{r}|\boldsymbol{q}) \, \text{for all } \boldsymbol{r} \in \mathcal{R}, \qquad (5.5)$$

---

[1]For simplicity, we omit the normalizing coefficient $1/|\boldsymbol{y}|$ from Eq. 3.3, which can be treated as a constant $1/2$ in our framework.

where $Z = \sum_{\boldsymbol{r} \in \mathcal{R}} \exp \left( \frac{1}{\beta} R(\boldsymbol{r}) \right) \pi_{\boldsymbol{\theta}_{\text{ref}}} (\boldsymbol{r}|\boldsymbol{q})$ *is the normalizing coefficient, and* $R(\boldsymbol{r})$ *denotes the reasoning path reward. Specifically:*

- *For RLVR,* $R(\boldsymbol{r})$ *equals the success rate of reasoning pattern* $\boldsymbol{r}$, *i.e.,* $R_{\text{RLVR}}(\boldsymbol{r}) = p^*(\boldsymbol{r})$.

- *for RLIF,* $R(\boldsymbol{r})$ *equals to the confidence for the reasoning pattern* $\boldsymbol{r}$ *for final answer distribution, i.e.,* $R_{\text{RLIF}}(\boldsymbol{r}) = -\frac{1}{|\mathcal{A}|} \sum_{\boldsymbol{a} \in \mathcal{A}} \log(\pi_\theta(\boldsymbol{a}|q, \boldsymbol{r}))$.

Proposition 5.2 characterizes the optimal solutions for RLVR and RLIF. Assuming $\pi_{\text{opt}}$ resides within the general autoregressive policy parameterized space (which always holds for both LLMs and the tabular policies discussed later), the RLVR-optimized model increases the probability of a reasoning pattern $\boldsymbol{r}$ when the its probability to deduct the correct answer $p^*(\boldsymbol{r})$ is high enough such that $\exp(\frac{1}{\beta} p^*(\boldsymbol{r}))/Z > 1$. In contrast, for RLIF, the formulation of $R_{\text{RLIF}}(\boldsymbol{r})$ just consider the confidence of the final answer distribution, fails to distinguish between correct answer $\boldsymbol{a}^*$ and incorrect alternatives. Consequently, while optimal policy under RLVR could consistently leading to better performance, the optimal policy for RLIF does not guarantee improved accuracy over the base model.

Moreover, since $\beta$ is typically small (e.g., 0.001) in practice Rafailov et al. (2023); Bai et al. (2022); Zeng et al. (2025), as $\beta \to 0$, we have:

$$\pi_{\text{opt}}(\boldsymbol{r}|\boldsymbol{q}) = \lim_{\beta \to 0} \frac{1}{Z} \exp \left( \frac{1}{\beta} R(\boldsymbol{r}) \right) \pi_{\boldsymbol{\theta}_{\text{ref}}}(\boldsymbol{r}|\boldsymbol{q}) = \begin{cases} 1 & \boldsymbol{r} = \arg\max_{\boldsymbol{r}} R(\boldsymbol{r}) \\ 0 & \text{otherwise} \end{cases}, \qquad (5.6)$$

In this limit, the policy converges to a deterministic strategy that always selects the reasoning pattern $\boldsymbol{r}$ with highest $R(\boldsymbol{r})$, regardless of how the initialized reference model chooses reasoning patterns. However, due to the non-convexity of the optimization landscape, the dynamics of how RLVR reliably finds high-reward patterns and why RLIF may exhibit early performance improvements remain unclear. To address this, we next analyze the training dynamics of both methods under a tabular policy (Eq. 5.3), providing further insight into the RL training process.

## 5.3 TRAINING DYNAMIC ANALYSIS FOR RLVR

Our experiments demonstrate that RLVR improvements heavily depend on the capacity of the base model. Here, we focus on the probability of the *optimal reasoning pattern*, i.e., the pattern $\boldsymbol{r}^*$ with the highest success rate for reaching the correct answer $\boldsymbol{a}^*$ $\boldsymbol{r}^* = \arg\max_{\boldsymbol{r}} p^*(\boldsymbol{r})$. We reveal two distinct regimes in the training dynamics of RLVR via gradient flow. In the first regime, the probability of the optimal reasoning pattern $\pi_\theta(\boldsymbol{r}^*|\boldsymbol{q})$ steadily increases until convergence to 1. In the second regime, the model initially experiences an entanglement stage, where a suboptimal reasoning pattern $\boldsymbol{r}' \neq \boldsymbol{r}^*$ hinder the optimizing process for the optimal reasoning pattern. After this entanglement stage, the model eventually transitions to the dynamics of the first regime and converges to the optimal reasoning pattern $\boldsymbol{r}^*$.

**Theorem 5.3** (Regime 1: Sufficient Condition for Efficient Convergence). *Consider the RLVR (Eq 3.2) with $\beta = 0$ for optimizing objective Eq 3.1, using a tabular policy (Eq 5.3) with Assumption 5.1 holds. Let $\boldsymbol{r}^*$ be the optimal reasoning pattern, if the overall accuracy of the initialized model $\pi_{\boldsymbol{\theta}_{\text{ref}}}$, defined as $\text{ACC}_{\boldsymbol{\theta}_{\text{ref}}} = \sum_{\boldsymbol{r} \in \mathcal{R}} \pi_{\boldsymbol{\theta}_{\text{ref}}}(\boldsymbol{r}|\boldsymbol{q}) p^*(\boldsymbol{r})$, satisfies:*

*(Regime 1)* $$\text{ACC}_{\boldsymbol{\theta}_{\text{ref}}} > p^*(\boldsymbol{r}) \text{ for all } \boldsymbol{r} \in \mathcal{R}, \boldsymbol{r} \neq \boldsymbol{r}^*, \qquad (5.7)$$

*then for any $\epsilon > 0$, there exists $T_1 = \mathcal{O}(\frac{1}{\epsilon})$ such that for $t > T_1$, we have $1 - \pi_{\boldsymbol{\theta}(t)}(\boldsymbol{r}^*|\boldsymbol{q}) < \epsilon$.*

In Theorem 5.3, we consider a case where the base reference model is sufficiently strong such that its overall accuracy exceeds the success rate of all non-optimal reasoning patterns (Eq 5.7). In this case, RLVR can efficiently guide the model to select the optimal reasoning pattern at rate $\mathcal{O}(1/\epsilon)$, achieving a high overall accuracy (close to the success rate of the optimal reasoning pattern).

However, practical experience shows that in some scenarios, RLVR optimization can be challenging Zeng et al. (2025); Xie et al. (2025), which typically occurs when the base reference model is less powerful. We then consider the second regime, where the model initially experiences an entanglement stage, and the *suboptimal reasoning pattern with the second-highest success rate*, defined as $\boldsymbol{r}' = \arg\max_{\boldsymbol{r}, \boldsymbol{r} \neq \boldsymbol{r}^*} p^*(\boldsymbol{r})$, slows down the optimization process for the optimal reasoning pattern:

**Theorem 5.4** (Regime 2: Slow Convergence for optimal reasoning pattern). *Consider the RLVR (Eq 3.2) with $\beta = 0$ for optimizing objective Eq 3.1, using a tabular policy (Eq 5.3) with Assumption 5.1 holds. Let $\boldsymbol{r}^*$ and $\boldsymbol{r}'$ be the optimal and second optimal reasoning patterns, if the overall accuracy of the initialized model satisfies:*

$$(Regime\ 2) \qquad p^*(\boldsymbol{r}') > \mathrm{ACC}_{\boldsymbol{\theta}_{\mathrm{ref}}} > p^*(\boldsymbol{r})\ \text{for all}\ \boldsymbol{r} \in \mathcal{R}/\{\boldsymbol{r}^*, \boldsymbol{r}'\}, \qquad (5.8)$$

*then there exists:*

$$T_0 = \frac{1}{2 - 2\pi_{\boldsymbol{\theta}_{\mathrm{ref}}}(\boldsymbol{r}'|\boldsymbol{q})} \left( (C_1 \cdot \gamma_{\pi_{\mathrm{ref}}})^{2C_2 \cdot \gamma_{\pi_{\mathrm{ref}}}} - 1 \right),\ \text{where}\ \gamma_{\pi_{\mathrm{ref}}} := \sum_{\boldsymbol{r} \in \mathcal{R}/\{\boldsymbol{r}'\}} \frac{\pi_{\boldsymbol{\theta}_{\mathrm{ref}}}(\boldsymbol{r}|\boldsymbol{q})}{\pi_{\boldsymbol{\theta}_{\mathrm{ref}}}(\boldsymbol{r}^*|\boldsymbol{q})} \qquad (5.9)$$

*with constants $C_1, C_2$ depending on the success rates of reasoning patterns, such that for we can guarantee the model transform from regime 2 (Eq 5.8) to regime 1 (Eq 5.7), i.e., $\mathrm{ACC}_{\boldsymbol{\theta}(t)} > p^*(\boldsymbol{r}), \forall \boldsymbol{r} \in \mathcal{R}, \boldsymbol{r} \neq \boldsymbol{r}^*$ for $t \geq T_0$.*

In Theorem 5.4, we consider a special regime where only the success rates of the optimal and suboptimal reasoning patterns exceed the overall accuracy. We consider this case for ease of theoretical analysis and believe it can be extended to more general settings where *at least* two reasoning patterns are allowed to achieve higher success rates than the average, albeit with more complicated theoretical analysis.

Additionally, the critical insight of Theorem 5.4 is that RLVR may require $T_0$ time steps to ensure that the overall success rate exceeds the success rate of the suboptimal reasoning pattern, i.e., to reach the regime discussed in Theorem 5.3 where the selection probability of the optimal reasoning pattern is sufficiently large. While Theorem 5.3 shows that the convergence time $T_1$ is polynomial in $1/\epsilon$, the time step $T_0$ in Theorem 5.4 may grow super-exponentially with respect to $\gamma_{\pi_{\mathrm{ref}}}$—the ratio between the total success rate through $\mathcal{R}/\{\boldsymbol{r}'\}$ and the success rate through $\boldsymbol{r}^*$. Clearly, when the base model assigns a very small selection probability to the optimal reasoning pattern, we may have a very large $\gamma_{\pi_{\mathrm{ref}}}$, which leads to a prohibitively large $T_0$. Consequently, it can take an extremely long training period for the model to select the optimal reasoning pattern with a reasonably large probability, which we refer to as the entanglement stage.

We provide additional experiments and case studies to further illustrate these two regimes in Sec 6.

## 5.4 THEORETICAL EXPLANATION FOR RLIF

In Proposition 5.2, we reveal that RLIF, which only considers the confidence of the final answer distribution, fails to distinguish between the correct answer $\boldsymbol{a}^*$ and incorrect alternatives. This indicates that RLIF may eventually lead to performance degradation compared to the base model, which aligns well with our empirical observations. However, previous studies have demonstrated that RLIF can improve model performance without external rewards Agarwal et al. (2025); Zhao et al. (2025b). Here, we provide a theoretical explanation for why RLIF can improve the performance of a well-trained LLM at the initial training stage.

First, we consider a well-trained LLM satisfying the following assumption:

**Assumption 5.5.** *For the base model, the correct answer $\boldsymbol{a}^*$ has the highest probability across all possible answers, i.e., $\arg\max_{\boldsymbol{a} \in \mathcal{A}} \sum_{\boldsymbol{r} \in \mathcal{R}} \pi_{\mathrm{base}}(\boldsymbol{a}|q, \boldsymbol{r})\pi_{\mathrm{base}}(\boldsymbol{r}|q) = \boldsymbol{a}^*$.*

Since majority voting Wang et al. (2022) has proven to be an effective method for improving model performance, and such technique tends to choose the answer with the highest probability Wu et al. (2025), such improvements indicate a well-trained LLMs tend assign a higher probability to the correct answer $\boldsymbol{a}^*$ than to incorrect answers, Assumption 5.5 is likely to hold for modern LLMs in many problems. Under this constraint, we consider the case that $|\mathcal{A}| = 2$, the success rate for each reasoning path follows a uniform distribution and the reasoning pattern selection is high-entropy[2]. In this case, we analyze the overall accuracy dynamics at $t = 0$, yielding the following theorem:

**Theorem 5.6** (RLIF Increases Overall Accuracy at Initialization). *Consider the RLIF (Eq 5.1) with $\beta = 0$ for optimizing objective Eq 3.1, using a tabular policy (Eq 5.3) with Assumption 5.1 and 5.5 hold, $\pi_{\boldsymbol{\theta}_{\mathrm{base}}}(\boldsymbol{r}|q) = \frac{1}{|\mathcal{R}|}$ for all $\boldsymbol{r} \in \mathcal{R}$, $|\mathcal{A}| = 2$ and success rate for each reasoning path follows a uniform distribution $p^*(\boldsymbol{r}) \sim U[0, 1]$. Then when $|\mathcal{R}| \to +\infty$, the following holds:*

---

[2]Previous work (Wang et al., 2025a) reveals that RL primarily optimizes high-entropy tokens. Here, we consider the highest entropy case: $\pi_{\boldsymbol{\theta}_{\mathrm{base}}}(\boldsymbol{r}|q) = 1/|\mathcal{R}|$ for all $\boldsymbol{r} \in \mathcal{R}$.

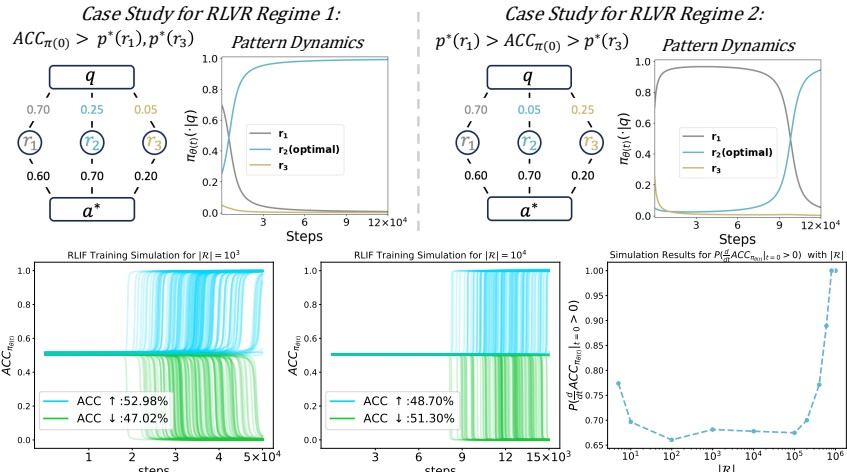

Figure 2: Case studies for RLVR (upper) and training simulations for RLIF (bottom). The upper panels demonstrate two distinct training regimes in RLVR: rapid convergence (left) and entanglement phase (right). The bottom panels show the probabilities of model convergence to different states (left), and the probability of initial performance gains (right) for RLIF across varying $|\mathcal{R}|$.

1. *The accuracy derivative at $t = 0$ is positive with probability $1$, i.e. $P(\frac{d}{dt}ACC_{\theta(t)}\big|_{t=0} > 0) = 1$.*
2. *With probability $p = 0.5$, that $\arg\max_{\boldsymbol{r}} R_{\mathrm{RLIF}}(\boldsymbol{r}) = \arg\min_{\boldsymbol{r}} p^*(\boldsymbol{r})$.*

The first result in Theorem 5.6 shows that overall accuracy increases at initialization, offering a theoretical explanation for RLIFs early performance gains. However, as training progresses, Proposition 5.2 implies that when $\beta = 0$, the policy converges deterministically to the reasoning pattern $\boldsymbol{r}$ that maximizes $R_{\mathrm{RLIF}}(\boldsymbol{r})$. The second result in Theorem 5.6 reveals that, with probability $0.5$, this maximizing pattern coincides with the one that minimizes the success rate: $\arg\max_{\boldsymbol{r}} R_{\mathrm{RLIF}}(\boldsymbol{r}) = \arg\min_{\boldsymbol{r}} p^*(\boldsymbol{r})$. In other words, there is a 50% chance the model converges to the least accurate reasoning path. Consequently, while RLIF initially improves model performance (as the derivative of accuracy is positive), continued training may cause the model to converge to states that favor reasoning paths with very low accuracy, ultimately resulting in performance worse than the base model.

## 6  CASE STUDIES AND NUMERICAL SIMULATIONS

To further validate and interpret our theoretical findings, we present case studies and training simulations to illustrate our theorems. For RLVR, we demonstrate two distinct training regimes discussed in Theorems 5.3 and 5.4, as illustrated in Fig. 2 (upper). For RLIF, we conduct multiple training simulations where each reasoning path's success rate follows a uniform distribution, validating our analysis of RLIF's behavior during initial training steps and at convergence, as shown in Fig. 2 (bottom).

### 6.1  CASE STUDIES FOR RLVR

In Section 5.3, we identified two distinct regimes in RLVR's training dynamics. In the first regime (Eq 5.7), the probability of selecting the optimal reasoning pattern $\pi_{\boldsymbol{\theta}}(\boldsymbol{r}^*|\boldsymbol{q})$ monotonically increases until convergence. In the second regime (Eq 5.8), the model undergoes an initial entanglement phase and requires a substantially longer training period before transitioning to the dynamics of the first regime and converging to the optimal reasoning pattern $\boldsymbol{r}^*$. We examine both regimes:

- As demonstrated in Fig. 2 (upper left), when the reference model satisfies the conditions in Eq 5.7, where the overall accuracy exceeds the success rates of all non-optimal reasoning patterns, the model converges rapidly to the optimal reasoning pattern.
- In the second case, corresponding to Theorem 5.4, when a suboptimal reasoning pattern $\boldsymbol{r}'$ achieves a higher success rate than the initial overall accuracy, the model first experiences an entanglement phase. A transition period $T_0$ (defined in Theorem 5.4) must elapse before the model enters the rapid optimization phase characteristic of case 1. Fig. 2 (upper right) illustrates a scenario where

$\gamma_{\pi_{\text{ref}}}$ (defined in Eq 5.9) is large ($\gamma_{\pi_{\text{ref}}} = 6$, as $(0.25 + 0.05)/0.05 = 6$). In such cases, the transition time $T_0$, which scales with $\gamma_{\pi_{\text{ref}}}^{\gamma_{\pi_{\text{ref}}}}$, becomes prohibitively long, significantly delaying convergence to the optimal reasoning pattern. The pattern dynamics in Fig. 2 clearly demonstrate such entanglement stage, align well with our theoretical result.

## 6.2 Numerical Simulations For RLIF

In Section 5.4, we demonstrated that under Assumption 5.5 and the base model distribution specified in Theorem 5.6, as $|\mathcal{R}| \to \infty$, the accuracy derivative at $t = 0$ is positive with probability 1, while there exists a 50% probability of convergence to the least accurate reasoning path. Our simulations validate these findings. Fig. 2 (bottom left) presents RLIF training simulations across varying $|\mathcal{R}|$. The results confirm that the probability of convergence to an improved state ($ACC \uparrow$) approximately equals the probability of convergence to a degraded state ($ACC \downarrow$), validating the second result in Theorem 5.6. Additionally, our examination of initial training step performance, shown in Fig. 2 (bottom right), demonstrates that for large $|\mathcal{R}|$, the probability of initial performance improvement approaches 1, supporting our first result in Theorem 5.6.

## 7 Conclusions and Limitations

This work analyzes reinforcement learning dynamics in LLMs through both empirical investigations and theoretical frameworks. We develop mathematical analyses for two representative reward mechanisms (RLVR and RLIF) and validate our findings through case studies and simulations. Our analysis has certain limitations that warrant further investigation: the interpretability of LLM-identified reasoning patterns needs additional validation, our theoretical framework could be extended to handle more complex real-world reasoning scenarios, and the current analysis could be generalized beyond specific base model assumptions.

## 8 Acknowledgments

We would like to thank the anonymous reviewers and area chairs for their helpful comments. We acknowledge the support from NSFC 62306252, Hong Kong ECS award 27309624, Guangdong NSF 2024A1515012444, and the central fund from HKU IDS.

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

## A    THE USE OF LARGE LANGUAGE MODELS

In this paper, we utilized LLMs to perform grammatical corrections and to generate code for data visualization and model training.

## B    ADDITIONAL EXPERIMENTS

### B.1    PATTERN DISTRIBUTION DYNAMICS ON RL-ENHANCED QWEN2.5-3B

In section 4.2, we sample Number Theory tasks from MATH to analyze the dynamics of the reasoning patterns. We also sample Geometry tasks for dynamic analysis (Figure 3). During RLVR, the model gradually adopts the reasoning patterns with higher accuracy, whereas the patterns selection for RLIF is not stable. This result is consistent with the empirical and theoretical results presented in the main paper.

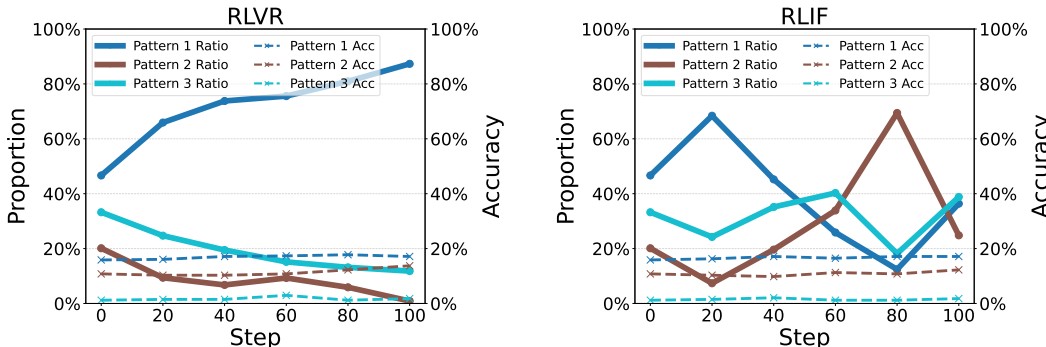

Figure 3: Dynamics of the patterns distribution and corresponding accuracy of RL-enhanced models on Geometry tasks. In RLVR, the model tends to choose the pattern with the highest accuracy (Pattern 1), but RLIF is not stable.

### B.2    QWEN-2.5-7B-INSTRUCT VS. QWEN-2.5-7B-SIMPLERL-ZOO

In order to verify whether our findings are applicable to different models, we conduct supplementary experiments to validate the experimental insights and theoretical conclusions (Figure 4). For these experiments, we use Qwen-2.5-7B-Instruct (Yang et al., 2024) as the base model and compare it with Qwen-2.5-7B-SimpleRL-Zoo (Zeng et al., 2025), a open-source variant enhanced with RLVR for mathematical reasoning. Our evaluation spans diverse mathematical domains, including number theory, geometry, algebra, calculus, counting and probability, using challenging problems sampled from the MATH dataset (Hendrycks et al., 2021). We further extend our analysis to include complex mathematical problems from AMC23 (2023., 2023). The results consistently support our earlier findings: RLVR enhancement shows an increase in the frequency of high-accuracy reasoning patterns, while less effective patterns appear less frequently. Detailed task-specific analyses are provided in Appendix C.3.

Through reasoning pattern analysis across various tasks, we observe that patterns with the highest success rates consistently become more prevalent after RLVR enhancement, reinforcing our findings from Section 4 and Section 5. Notably, in tasks such as Algebra and AMC23-19 in Figure 4, we observe consistent success rates across individual reasoning patterns, which not only aligns with our previous observations but also provides empirical support for Assumption 5.1 in our theoretical analysis.

### B.3    QWEN-2.5-32B-INSTRUCT VS. QWQ-32B

To further verify our theory, we conduct experiments on a larger model, QwQ-32B (Team, 2025). This model is based on Qwen-2.5-32B-Instruct (Yang et al., 2024) and greatly enhances the reasoning ability through RLVR. We test the reasoning patterns shift of these two models on four tasks, i.e.

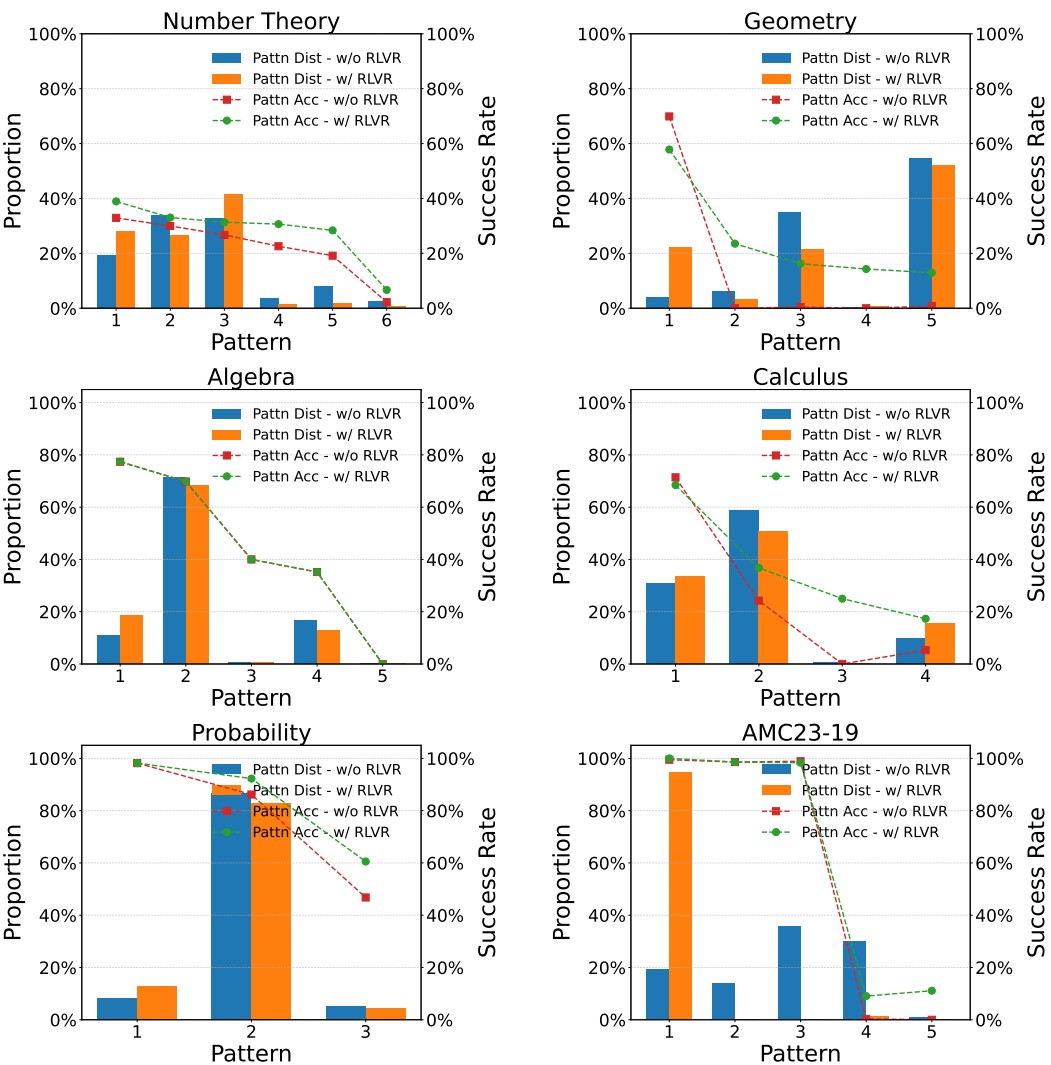

Figure 4: Evaluation results for reasoning pattern and corresponding success rate of model with/without RLVR enhancement for varying additional tasks, which are aligned well with our experimental insights and theoretical conclusions. The bar (Pattern Dist) represents the proportion of a certain pattern in all patterns. The dot (Pattern Acc) represents the accuracy or success rate corresponding to the pattern.

number theory and geometry tasks from the MATH dataset (Hendrycks et al., 2021), task from AIME 2024 (Codeforces), and task from OlympiadBench (He et al., 2024).

We can see that most of the reasoning patterns of QwQ-32B correspond to the most accurate patterns of Qwen-2.5-32B, which is aligned well with our theoretical results. We do not give the accuracy of QwQ because we use the API of the models for testing and can not extract the CoT data to allow the base model to continue to generate answers. In this way, it is impossible to obtain the accuracy of the reasoning patterns (rather than the accuracy of the model itself). For details, see the "Accuracy Analysis" in Appendix C.2.

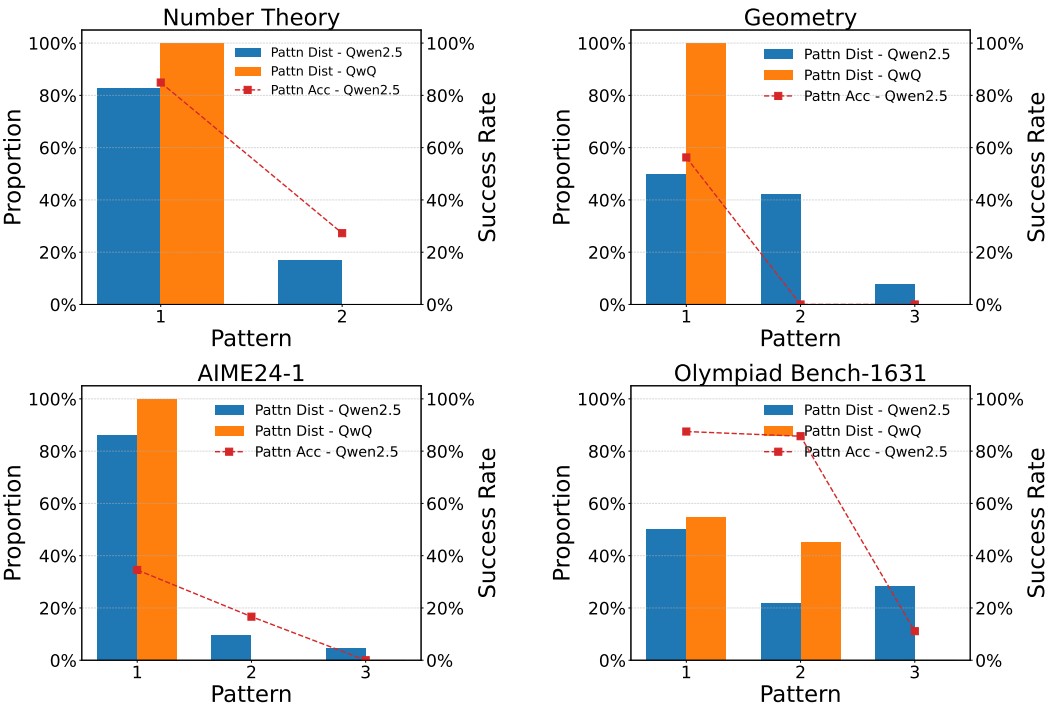

Figure 5: Evaluation results on larger models, Qwen-2.5-32B and QwQ-32B. QwQ's reasoning patterns converge to the most accurate pattern of Qwen-2.5, which demonstrates the applicability of our theory to larger-scale models.

## C EXPERIMENTAL DETAILS

### C.1 EXPERIMENTAL SETUP

**Entire Training Procedures:** We first select Qwen2.5-3B as the base model and train it on the MATH dataset, using both RLVR and RLIF. We adopt the verl (Sheng et al., 2024) framework and ensure that all parameter settings are identical to the example on the MATH dataset provided by the verl framework. For RLIF, we only modified the reward calculation method. By setting different random seeds, we obtain three different training curves for each method. All our settings are exactly the same as those in the Verl framework using the GRPO algorithm. The temperature is set to 1 during training. The learning rate is set to 1e-6. The train_batch_size is set to 1024 while the ppo_mini_batch_size is set to 256. In group sampling, the number of rollouts for a single prompt is set to 5. We disable top-k and set top-p to 1.

**Token-level Analysis:** We use Qwen2.5-3B and the above RL-enhanced models for this experiment. For GSM8K and MATH, we randomly sample 400 questions from their test sets respectively. For AIME24, we use its full dataset. We first sample the corresponding answer from the base model for each question. Next, we concatenate this answer to the question itself and input it into the RL-enhanced model as a new prompt. We use the interface in the OpenAI library to calculate the

probability of each token in the prompt and its ranking among the probabilities of all possible tokens at the current position. By comparing whether the ranking of each token has changed before and after RL, we can calculate the proportion of tokens whose ranking has changed.

**Reasoning-pattern Level Analysis:** For this task, we conduct two types of experiments. We first study the dynamics of the patterns distribution during training using previously trained models, namely Qwen2.5-3B and RL-enhanced Qwen2.5-3B. We evaluate every 20 steps. To examine the applicability of our conclusions on different base models and larger models, we test the open-source model Qwen-2.5-7B-SimpleRL-Zoo, QwQ-32B, and their corresponding base models. Therefore, we can only examine the changes in the patterns distribution of model outputs before and after RL, but cannot examine the dynamics of the patterns distribution during the training process.

## C.2 REASONING PATTERNS ANALYSIS PIPELINE

**Model Inference:** We first randomly sample questions of different task types (such as Geometry, Algebra, etc.) and different difficulty levels (Level 4, Level 5) from datasets including MATH. For each question, our model samples 1024 answers in the patterns distribution dynamic task, and samples answers ranging from 64 to 1024 in the task that only considers the distribution change before and after RL, depending on the task category. These responses will be used for subsequent pattern extraction and classification.

**Reasoning Patterns Extraction:** We sample a subset of responses of the base model and use GPT-4o (Hurst et al., 2024) API to summarize patterns categories from these samples. Full prompt we used for reasoning patterns extraction is given in Appendix E.1. To ensure the accuracy of patterns extraction, we set the temperature of GPT-4o's API to 0.

**Responses Classification:** We again employ GPT-4o to classify all responses according to the identified pattern categories, also setting the temperature to 0. Full prompt we used for responses classification is given in Appendix E.2.

**Accuracy Analysis:** In order to only consider the impact of the patterns distribution on accuracy, we construct new prompts by concatenating the original prompt with partial responses from the base model, assuming these partial responses sufficiently represent specific reasoning patterns. This ensures that the reasoning process is fixed, which makes it easier for us to determine whether the change in model accuracy depends only on the change in the distribution of the reasoning pattern, rather than the change in the reasoning process itself. Specifically, we remove the sentence containing the final answer from the base model's response, add the sentence before it to the end of the question, and input it as a new prompt to the RL-enhanced model. The RL-enhanced model will continue to predict the answer. We will examine the accuracy of the predicted answer as the accuracy of the reasoning pattern.

## C.3 DETAILS FOR EACH TASK

Below we detail the experimental procedures for each task, including the task descriptions and examples of reasoning patterns (Tabel 2). Example task prompts are provided in Appendix E.3.

**Number Theory:** This task presents models with problems involving coin distribution across multiple bags. Initially, bags contain equal numbers of coins. After receiving additional coins and redistributing them equally, the total must exceed a specified value while maintaining equal distribution. Models must determine the minimum initial coin count per bag. We derived this task from a level 5 MATH dataset problem (id: test/number_theory/1055) (Hendrycks et al., 2021), creating 32 variants by adjusting parameters like bag count and coin totals as our evaluation dataset. We employ one-shot prompting with a simple, unrelated example to guide answer formatting using "boxed" notation without influencing reasoning approaches. For evaluation, we first choose 20 questions from our evaluation dataset with 4 responses each for reasoning pattern extraction. For each question we sample 64 responses. We then categorize all responses by the extracted reasoning patterns, allowing us to compare changes in the distribution of models reasoning patterns before and after RL.

| Task | Reasoning Patterns Examples | Common Elements / Key Words |
|---|---|---|
| Number Theory | Modular Congruence with Coefficient Simplification | Uses modular arithmetic to express divisibility conditions. |
| | Inequality-Driven Search for Minimal Solution | Often involves substituting back to compute the total coins. |
| Geometry | Systematic Inequality Application | Explicitly lists and solves each of the three inequalities. |
| | Verification of Scalene Condition | Adding an extra layer of validation to verifies the third side length. |
| Algebra | Iterative Floor Division with Leftover Tracking. | Uses floor division to compute new cans per step. |
| | Recursive Recycling with Aggregated Leftovers. | Combines leftovers with newly produced cans before recycling. |
| Calculus | Direct Simplification and Principal Value Matching | Assumes inputs fall within the principal range. |
| | Interval Analysis with Case Splitting | Splits the whole domain into different intervals. |
| Counting and Probability | Direct Probability Setup and Quadratic Solution | Explicitly calculates combinations for total and favorable outcomes. |
| | Early Simplification and Cross-Multiplication | Early cross-multiplication to eliminate denominators. |
| AMC23-Q19 | Prime Factorization and Simplification | Counting Digits in Large Numbers. |
| | Rewriting $(8^5)$ as $(2^{15})$ $(15^5)$ as $(3^5 \cdot 5^5)$. | Final step of counting digits: $(3 \text{ (from 243)} + 15 \text{ (zeros)} = 18)$. |
| AIME24-Q1 | Direct Equation Setup and Elimination | Straightforward and relies on algebraic manipulation. |
| | Alternative Equation Formulation and Solving | Expressing variables in terms of others early on. |
| OlympiadBench-Q1631 | Direct Calculation and Empirical Testing | Relying on direct computation and empirical verification. |
| | Factorization-Based Reasoning | Using algebraic factorization to argue certain terms $y_n$. |

Table 2: Reasoning Patterns Examples for Varying Tasks.

**Geometry:** For the geometry task, we will give the model the lengths of two sides of a scalene triangle and ask how many different integer centimeters the length of the third side can be. This task comes from a level 4 geometry problem in MATH (id: test/geometry/1046). Our evaluation follows the same pipeline as in Number Theory: we construct a synthetic dataset with 32 questions, sample 20 questions and select 4 responses each from the base model (Qwen2.5-7b-Instruct) for reasoning pattern extraction, and then we compare the responses for model with and without RLVR enhancement with all questions in our evaluation dataset with 64 responses each for pattern analysis.

**Algebra:** The algebra task we use is a can recycling problem. We first have a certain number of old cans. It is pre-defined that $n$ old cans can be recycled into a new can. The question is how many cans can be produced in the end. The difficulty of the problem is that in each step of the iterative calculation, there may be extra cans that cannot be divided evenly. These cans may eventually be combined together for further recycling. The template for this task comes from a level 4 algebra problem in MATH (id: test/algebra/2768). For all the following supplementary tasks starting from this task, we adopt the same settings as the previous tasks, including the data set size, number of samples, etc.

**Calculus:** Our calculus task is simple and straightforward. We will present a trigonometric equation in a single variable, the domain of that variable, and ask the model to determine the number of solutions to the equation. Although this problem is simple, it can demonstrate the model's basic ability in calculus problems. Its prototype is a level 4 pre-calculus problem in MATH (id: test/precalculus/1140).

**Counting and Probability:** We also study the performance of the model on the counting and probability task. There are white balls and black balls. We will randomly sample two balls from these balls and give the probability that one of the two balls drawn is black and the other is white. We also provide the number of balls of a certain color and hope that the model can calculate the minimum number of balls of another color. This is a probability theory task, which comes from a level 4 counting and probability problem in MATH (id: test/counting_and_probability/79).

**AMC23 Question:** Here we choose question 19 in AMC23 (2023., 2023), for reasoning pattern extraction, we randomly sample 48 responses, and then sample 1024 response for reasoning pattern analysis.

### C.4 Details for Experiments on QwQ-32B and Qwen-2.5-32B

We conduct experiments on four tasks. Due to the limitations of using the API for testing, we randomly select one question per task as input, sample 64 responses from each of the two models, and directly perform reasoning patterns extraction and classification from these answers. Specifically, for the two tasks of the MATH dataset, we choose question 1055 from the number theory task and question 1046 from the geometry task. For the remaining two tasks, we question problem 1 from AIME24 and question 1631 from Olympiad Bench. Examples of patterns are shown in Table 2.

## D Deferred Proofs

In this appendix, we provide proofs for our main theoretical results: Proposition 5.2, Theorems 5.3, Theorems 5.4 and Theorems 5.6.

### D.1 Proof of Proposition 5.2

*Proof of Proposition 5.2.* In this proof, we utilize the proof techniques in Rafailov et al. (2023), recall that the optimization objective of RL is

$$\phi_{\mathrm{RL}}(\boldsymbol{\theta}) = \mathbb{E}_{\boldsymbol{x}\sim\mathcal{D},\boldsymbol{y}\sim\pi_{\boldsymbol{\theta}}(\boldsymbol{y}|\boldsymbol{x})}\left[r_{\phi}(\boldsymbol{x},\boldsymbol{y})\right] - \beta\mathbb{D}_{\mathrm{KL}}\left[\pi_{\boldsymbol{\theta}}(\boldsymbol{y}\mid\boldsymbol{x}) \parallel \pi_{\mathrm{ref}}(\boldsymbol{y}\mid\boldsymbol{x})\right], \quad\quad \text{(D.1)}$$

Under our framework, as state in section 5.1, we consider a policy conduct reasoning by first sample $\boldsymbol{r}_i \in \mathcal{R}$ based on $\pi_{\boldsymbol{\theta}}(\boldsymbol{r}_i|\boldsymbol{q})$ and then provide the final answer $\boldsymbol{a} \in \mathcal{A}$ by $\pi_{\boldsymbol{\theta}}(\boldsymbol{a}|\boldsymbol{r}_i)$, the reward $\phi_{\mathrm{RL}}(\boldsymbol{\theta})$

can be written as

$$\phi_{\mathrm{RL}}(\boldsymbol{\theta}) = \mathbb{E}_{\boldsymbol{y}\sim\pi_{\boldsymbol{\theta}}(\boldsymbol{r}|q)} \left[ \sum_{\boldsymbol{a}\in\mathcal{A}} \pi_{\mathrm{ref}}(\boldsymbol{a} \mid \boldsymbol{r}) r_{\phi}([\boldsymbol{r}, \boldsymbol{a}], q) \right] - \beta\mathbb{D}_{\mathrm{KL}}\left[\pi_{\boldsymbol{\theta}}(\boldsymbol{y} \mid \boldsymbol{x}) \parallel \pi_{\mathrm{ref}}(\boldsymbol{y} \mid \boldsymbol{x})\right], \quad \text{(D.2)}$$

Let $R(\boldsymbol{r}) = \sum_{\boldsymbol{a}\in\mathcal{A}} \pi_{\mathrm{ref}}(\boldsymbol{a} \mid \boldsymbol{r}) r_{\phi}([\boldsymbol{r}, \boldsymbol{a}], q)$, then

$$\begin{aligned}
\phi_{\mathrm{RL}}(\boldsymbol{\theta}) &= \mathbb{E}_{\boldsymbol{r}\sim\pi_{\boldsymbol{\theta}}(\cdot|\boldsymbol{q}), \boldsymbol{a}\sim\pi_{\boldsymbol{\theta}}(\cdot|\boldsymbol{r})} \left[ R(\boldsymbol{r}) - \beta\ln\left( \frac{\pi_{\boldsymbol{\theta}}(\boldsymbol{r}|\boldsymbol{q}) \cdot \pi_{\mathrm{ref}}(\boldsymbol{a}|\boldsymbol{r})}{\pi_{\mathrm{ref}}(\boldsymbol{r}|\boldsymbol{q}) \cdot \pi_{\mathrm{ref}}(\boldsymbol{a}|\boldsymbol{r})} \right) \right] \\
&= \mathbb{E}_{\boldsymbol{r}\sim\pi_{\boldsymbol{\theta}}(\cdot|\boldsymbol{q})} \left[ R(\boldsymbol{r}) - \beta\ln\left( \frac{\pi_{\boldsymbol{\theta}}(\boldsymbol{r}|\boldsymbol{q})}{\pi_{\mathrm{ref}}(\boldsymbol{r}|\boldsymbol{q})} \right) \right] \\
&= -\beta\mathbb{E}_{\boldsymbol{r}\sim\pi_{\boldsymbol{\theta}}(\cdot|\boldsymbol{q})} \left[ \ln\left( \frac{\pi_{\boldsymbol{\theta}}(\boldsymbol{r}|\boldsymbol{q})}{\frac{1}{Z}\exp\left(\frac{1}{\beta}R(\boldsymbol{r})\right)\pi_{\mathrm{ref}}(\boldsymbol{r}|\boldsymbol{q})} \right) - \ln Z \right] \\
&= -\beta\mathbb{D}_{\mathrm{KL}}\left[\pi^{*}(\boldsymbol{r}|\boldsymbol{q}) \parallel \pi_{\mathrm{ref}}(\boldsymbol{y} \mid \boldsymbol{x})\right] + \beta\ln Z.
\end{aligned}$$

Where the third equation is by Assumption 5.1 and $Z = \sum_{\boldsymbol{r}\in\mathcal{R}} \exp\left(\frac{1}{\beta}p^{*}(\boldsymbol{r})\right) \pi_{\boldsymbol{\theta}_{\mathrm{ref}}}(\boldsymbol{r}|\boldsymbol{q})$ is the partition constant that ensures

$$\pi^{*}(\boldsymbol{r}|\boldsymbol{q}) = \frac{1}{Z}\exp\left(\frac{1}{\beta}R(\boldsymbol{r})\right)\pi_{\boldsymbol{\theta}}(\boldsymbol{r}|\boldsymbol{q}),$$

is a valid probability distribution such that $\sum_{\boldsymbol{r}\in\mathcal{R}} \pi^{*}(\boldsymbol{r}|\boldsymbol{q}) = 1$. Since $Z$ is not a function of $\boldsymbol{r}$.

Therefore, maximizing the objective in Equation D.2 is equivalent to:

$$\begin{aligned}
&\max_{\pi} \mathbb{E}_{\boldsymbol{x}\sim\mathcal{D}, \boldsymbol{y}\sim\pi(\boldsymbol{y}|\boldsymbol{x})} \left[ r_{\phi}(\boldsymbol{x}, \boldsymbol{y}) \right] - \beta\mathbb{D}_{\mathrm{KL}}\left[\pi(\boldsymbol{y} \mid \boldsymbol{x}) \parallel \pi_{\mathrm{ref}}(\boldsymbol{y} \mid \boldsymbol{x})\right] \\
&= \min_{\pi} \beta\mathbb{D}_{\mathrm{KL}}\left[\pi^{*}(\boldsymbol{r}|\boldsymbol{q}) \parallel \pi_{\mathrm{ref}}(\boldsymbol{y} \mid \boldsymbol{x})\right] - \beta\ln Z.
\end{aligned}$$

By the properties of KL-divergence, we know that the optimal policy for the KL-constrained reward maximization objective satisfies:

$$\pi_{opt}(\boldsymbol{r}|\boldsymbol{q}) = \frac{1}{Z}\exp\left(\frac{1}{\beta}R(\boldsymbol{r})\right)\pi_{\boldsymbol{\theta}_{\mathrm{ref}}}(\boldsymbol{r}|\boldsymbol{q}) \text{ for all } \boldsymbol{r} \in \mathcal{R}.$$

specifically, for RLVR, $R_{\mathrm{RLVR}}(\boldsymbol{r}) = p^{*}$, for RLIF, we have

$$\begin{aligned}
R'(\boldsymbol{r}) &= -\left( \frac{1}{|\mathcal{R}|}\sum_{\boldsymbol{r}_i\in\mathcal{R}} \log\left(|\mathcal{R}| \cdot \pi_{\boldsymbol{\theta}}(\boldsymbol{r}_i|q)\right) + \frac{1}{|\mathcal{A}|}\sum_{\boldsymbol{a}_j\in\mathcal{A}} \log\left(|\mathcal{A}| \cdot \pi_{\boldsymbol{\theta}}(\boldsymbol{a}_j|q, \boldsymbol{r})\right) \right) \\
&= -\frac{1}{|\mathcal{A}|}\sum_{\boldsymbol{a}_j\in\mathcal{A}} \log\left(\pi_{\boldsymbol{\theta}}(\boldsymbol{a}_j|q, \boldsymbol{r})\right) - \underbrace{\left( \frac{1}{|\mathcal{R}|}\sum_{\boldsymbol{r}_i\in\mathcal{R}} \log\left(|\mathcal{R}| \cdot \pi_{\boldsymbol{\theta}}(\boldsymbol{r}_i|q)\right) + \log|\mathcal{A}| \right)}_{c},
\end{aligned}$$

where the second term is the same for all $\boldsymbol{r} \in |\mathcal{R}|$, for RLIF, the optimal policy can be written as

$$\begin{aligned}
\pi_{opt}(\boldsymbol{r}|\boldsymbol{q}) &= \frac{1}{Z}\exp\left(\frac{1}{\beta}R'(\boldsymbol{r})\right)\pi_{\boldsymbol{\theta}_{\mathrm{ref}}}(\boldsymbol{r}|\boldsymbol{q}) \\
&= \frac{\exp\left(\frac{1}{\beta}R'(\boldsymbol{r})\right)\pi_{\boldsymbol{\theta}_{\mathrm{ref}}}(\boldsymbol{r}|\boldsymbol{q})}{\sum_{\boldsymbol{r}_i\in\mathcal{R}} \exp\left(\frac{1}{\beta}(R'(\boldsymbol{r}_i))\right)\pi_{\boldsymbol{\theta}_{\mathrm{ref}}}(\boldsymbol{r}_i|\boldsymbol{q})} \\
&= \frac{\exp\left(\frac{1}{\beta}(R'(\boldsymbol{r}) - c)\right)\pi_{\boldsymbol{\theta}_{\mathrm{ref}}}(\boldsymbol{r}|\boldsymbol{q})}{\sum_{\boldsymbol{r}_i\in\mathcal{R}} \exp\left(\frac{1}{\beta}(R'(\boldsymbol{r}_i) - c)\right)\pi_{\boldsymbol{\theta}_{\mathrm{ref}}}(\boldsymbol{r}_i|\boldsymbol{q})}
\end{aligned}$$

So we can write $R_{\mathrm{RLIF}}(\boldsymbol{r}) = R'(\boldsymbol{r}) - c = -\frac{1}{|\mathcal{A}|}\sum_{\boldsymbol{a}_j\in\mathcal{A}} \log\left(\pi_{\boldsymbol{\theta}}(\boldsymbol{a}_j|q, \boldsymbol{r})\right)$. This concludes the proof of the theorem. $\square$

## D.2 Dynamics of Tabular Policy

Consider the RL objective (Eq 3.1), using a tabular policy (Eq 5.3) with Assumption 5.1 holds, the gradient dynamics of $\boldsymbol{\theta}$ can be computed as

$$
\begin{aligned}
\frac{d}{dt}\boldsymbol{\theta}(t) &= \nabla\phi_{\mathrm{RL}}(\boldsymbol{\theta}(t)) \\
&= \nabla\mathbb{E}_{\boldsymbol{r}\sim\pi_{\boldsymbol{\theta}}(\cdot|\boldsymbol{q})}\left[r_\phi^{KL}(\boldsymbol{r}|q)\right] \\
&= \sum_{\boldsymbol{r}\in\mathcal{R}}\left[r_\phi^{KL}(\boldsymbol{r}|q)\nabla\pi_{\boldsymbol{\theta}(t)}(\boldsymbol{r}|\boldsymbol{q})\right],
\end{aligned}
$$

Where $\pi_{\boldsymbol{\theta}(t)}(\boldsymbol{r}|\boldsymbol{q}) = \mathrm{softmax}(\boldsymbol{\theta}_{:,\boldsymbol{q}})_{\boldsymbol{r}}$, $\boldsymbol{\theta}_{:,\boldsymbol{q}} \in \mathbb{R}^{|\mathcal{V}|}$ is the column of $\boldsymbol{\theta}$ corresponding to $\boldsymbol{q}$, and $r_\phi(\boldsymbol{r}|q)^{KL}$ refers to the reward with KL divergence, so

$$
\begin{aligned}
\frac{\partial\boldsymbol{\theta}_{\boldsymbol{r}_i,\boldsymbol{q}}(t)}{\partial t} &= \sum_{\boldsymbol{r}_j\in\mathcal{R}}\left[r_\phi^{KL}(\boldsymbol{r}_j|q)\left(\nabla\mathrm{softmax}(\boldsymbol{\theta}_{:,\boldsymbol{q}})_{\boldsymbol{r}_j}\right)_{\boldsymbol{r}_i}\right] \\
&= \sum_{\boldsymbol{r}_j\in\mathcal{R}}\left[r_\phi^{KL}(\boldsymbol{r}_j|q)\left(-\pi_{\boldsymbol{\theta}(t)}(\boldsymbol{r}_j|\boldsymbol{q})\cdot\pi_{\boldsymbol{\theta}(t)}(\boldsymbol{r}_i|\boldsymbol{q})\right)\right] \\
&\quad + r_\phi^{KL}(\boldsymbol{r}_i|q)\cdot\pi_{\boldsymbol{\theta}(t)}(\boldsymbol{r}_i|\boldsymbol{q})
\end{aligned}
$$

where the overall accuracy is defined as $\mathrm{ACC}_{\boldsymbol{\theta}(t)} = \sum_{\boldsymbol{r}\in\mathcal{R}}\pi_{\boldsymbol{\theta}(t)}(\boldsymbol{r}|\boldsymbol{q})p^*(\boldsymbol{r})$. For $\pi_{\boldsymbol{\theta}(t)}(\boldsymbol{r}_i|\boldsymbol{q})$, we have

$$
\frac{d}{dt}\pi_{\boldsymbol{\theta}(t)}(\boldsymbol{r}_i|\boldsymbol{q}) = \sum_{\boldsymbol{r}_j\in\mathcal{R}}\frac{\partial\pi_{\boldsymbol{\theta}(t)}(\boldsymbol{r}_i|\boldsymbol{q})}{\partial\boldsymbol{\theta}_{\boldsymbol{r}_j,\boldsymbol{q}}(t)}\cdot\frac{\partial\boldsymbol{\theta}_{\boldsymbol{r}_j,\boldsymbol{q}}(t)}{\partial t}
$$

## D.3 Proof of Theorem 5.3

To proof Theorem 5.3, here we first prove that $\mathrm{ACC}_{\boldsymbol{\theta}(t)} \geq \mathrm{ACC}_{\boldsymbol{\theta}(0)}$ for $t \geq 0$ (Eq D.4), then we establish a lower bound for $\pi_{\boldsymbol{\theta}(t)}(\boldsymbol{r}^*|\boldsymbol{q})$ (Eq D.5), finally we derive the final bound for $t$ stated in Theorem 5.3.

*Proof of Theorem 5.3.* Setting $\beta = 0$, Eq D.3 becomes:

$$
\frac{\partial\boldsymbol{\theta}_{\boldsymbol{r}_i,\boldsymbol{q}}(t)}{\partial t} = \pi_{\boldsymbol{\theta}(t)}(\boldsymbol{r}_i|\boldsymbol{q})\cdot\left(p^*(\boldsymbol{r}_i) - \mathrm{ACC}_{\boldsymbol{\theta}(t)}\right). \tag{D.3}
$$

For $\pi_{\boldsymbol{\theta}(t)}(\boldsymbol{r}_i|\boldsymbol{q})$, we derive:

$$
\begin{aligned}
\frac{d}{dt}\pi_{\boldsymbol{\theta}(t)}(\boldsymbol{r}_i|\boldsymbol{q}) &= \sum_{\boldsymbol{r}_j\in\mathcal{R}}\frac{\partial\pi_{\boldsymbol{\theta}(t)}(\boldsymbol{r}_i|\boldsymbol{q})}{\partial\boldsymbol{\theta}_{\boldsymbol{r}_j,\boldsymbol{q}}(t)}\cdot\frac{\partial\boldsymbol{\theta}_{\boldsymbol{r}_j,\boldsymbol{q}}(t)}{\partial t} \\
&= \sum_{\boldsymbol{r}_j\in\mathcal{R}}\left(\mathrm{ACC}_{\boldsymbol{\theta}(t)} - p^*(\boldsymbol{r}_j)\right)\pi_{\boldsymbol{\theta}(t)}^2(\boldsymbol{r}_j|\boldsymbol{q})\pi_{\boldsymbol{\theta}(t)}(\boldsymbol{r}_i|\boldsymbol{q}) \\
&\quad + \left(p^*(\boldsymbol{r}_i) - \mathrm{ACC}_{\boldsymbol{\theta}(t)}\right)\pi_{\boldsymbol{\theta}(t)}^2(\boldsymbol{r}_i|\boldsymbol{q}),
\end{aligned}
$$

and for the accuracy $\text{ACC}_{\boldsymbol{\theta}(t)}$:

$$
\begin{aligned}
\frac{d}{dt}\text{ACC}_{\boldsymbol{\theta}(t)} &= \sum_{\boldsymbol{r}_i \in \mathcal{R}} p^*(\boldsymbol{r}_i) \cdot \frac{d}{dt}\pi_{\boldsymbol{\theta}(t)}(\boldsymbol{r}_i|\boldsymbol{q}) \\
&= \sum_{\boldsymbol{r}_i, \boldsymbol{r}_j \in \mathcal{R}} p^*(\boldsymbol{r}_i) \cdot \left(\text{ACC}_{\boldsymbol{\theta}(t)} - p^*(\boldsymbol{r}_j)\right) \pi^2_{\boldsymbol{\theta}(t)}(\boldsymbol{r}_j|\boldsymbol{q})\pi_{\boldsymbol{\theta}(t)}(\boldsymbol{r}_i|\boldsymbol{q}) \\
&\quad + \sum_{\boldsymbol{r}_i \in \mathcal{R}} \left(p^*(\boldsymbol{r}_i) - \text{ACC}_{\boldsymbol{\theta}(t)}\right) \pi^2_{\boldsymbol{\theta}(t)}(\boldsymbol{r}_i|\boldsymbol{q}) \\
&= \sum_{\boldsymbol{r}_j \in \mathcal{R}} \text{ACC}_{\boldsymbol{\theta}(t)} \cdot \left(\text{ACC}_{\boldsymbol{\theta}(t)} - p^*(\boldsymbol{r}_j)\right) \pi^2_{\boldsymbol{\theta}(t)}(\boldsymbol{r}_j|\boldsymbol{q}) \\
&\quad + \sum_{\boldsymbol{r}_i \in \mathcal{R}} \left(p^*(\boldsymbol{r}_i) - \text{ACC}_{\boldsymbol{\theta}(t)}\right) \pi^2_{\boldsymbol{\theta}(t)}(\boldsymbol{r}_i|\boldsymbol{q})p^*(\boldsymbol{r}_i) \\
&= \sum_{\boldsymbol{r}_i \in \mathcal{R}} \pi^2_{\boldsymbol{\theta}(t)}(\boldsymbol{r}_i|\boldsymbol{q}) \left(p^*(\boldsymbol{r}_i) - \text{ACC}_{\boldsymbol{\theta}(t)}\right)^2 \geq 0.
\end{aligned}
\tag{D.4}
$$

As $\frac{d}{dt}\text{ACC}_{\boldsymbol{\theta}(t)} \geq 0$, so $\text{ACC}_{\boldsymbol{\theta}(t)} > \text{ACC}_{\boldsymbol{\theta}(0)}$ holds for $t \geq 0$, for the optimal reasoning pattern $\boldsymbol{r}^*$, let $\boldsymbol{r}'$ the suboptimal reasoning pattern with the second highest success rate, defined as $\boldsymbol{r}' = \arg\max_{\boldsymbol{r}, \boldsymbol{r} \neq \boldsymbol{r}^*} p^*(\boldsymbol{r})$, we have:

$$
\begin{aligned}
\frac{d}{dt}\pi_{\boldsymbol{\theta}(t)}(\boldsymbol{r}^*|\boldsymbol{q}) &= \sum_{\boldsymbol{r}_j \in \mathcal{R}} \left(\text{ACC}_{\boldsymbol{\theta}(t)} - p^*(\boldsymbol{r}_j)\right) \pi^2_{\boldsymbol{\theta}(t)}(\boldsymbol{r}_j|\boldsymbol{q})\pi_{\boldsymbol{\theta}(t)}(\boldsymbol{r}^*|\boldsymbol{q}) \\
&\quad + \left(p^*(\boldsymbol{r}^*) - \text{ACC}_{\boldsymbol{\theta}(t)}\right) \pi^2_{\boldsymbol{\theta}(t)}(\boldsymbol{r}^*|\boldsymbol{q}) \\
&\geq \left(\text{ACC}_{\boldsymbol{\theta}(t)} - p^*(\boldsymbol{r}^*)\right) \pi^2_{\boldsymbol{\theta}(t)}(\boldsymbol{r}^*|\boldsymbol{q})\pi_{\boldsymbol{\theta}(t)}(\boldsymbol{r}^*|\boldsymbol{q}) \\
&\quad + \left(p^*(\boldsymbol{r}^*) - \text{ACC}_{\boldsymbol{\theta}(t)}\right) \pi^2_{\boldsymbol{\theta}(t)}(\boldsymbol{r}^*|\boldsymbol{q}) \\
&= \left(p^*(\boldsymbol{r}^*) - \text{ACC}_{\boldsymbol{\theta}(t)}\right) \pi^2_{\boldsymbol{\theta}(t)}(\boldsymbol{r}^*|\boldsymbol{q}) \left(1 - \pi_{\boldsymbol{\theta}(t)}(\boldsymbol{r}^*|\boldsymbol{q})\right) \\
&\geq \left(p^*(\boldsymbol{r}^*) - p^*(\boldsymbol{r}')\right) \pi^2_{\boldsymbol{\theta}(t)}(\boldsymbol{r}^*|\boldsymbol{q}) \left(1 - \pi_{\boldsymbol{\theta}(t)}(\boldsymbol{r}^*|\boldsymbol{q})\right)^2 \geq 0.
\end{aligned}
\tag{D.5}
$$

As $\frac{d}{dt}\pi_{\boldsymbol{\theta}(t)}(\boldsymbol{r}^*|\boldsymbol{q}) \geq 0$, $\pi_{\boldsymbol{\theta}(t)}(\boldsymbol{r}^*|\boldsymbol{q}) \geq \pi_{\boldsymbol{\theta}(0)}(\boldsymbol{r}^*|\boldsymbol{q})$ holds for $t \geq 0$, define $C = \left(p^*(\boldsymbol{r}^*) - p^*(\boldsymbol{r}')\right) \pi^2_{\boldsymbol{\theta}(0)}(\boldsymbol{r}^*|\boldsymbol{q})$:

$$
\frac{d}{dt}\pi_{\boldsymbol{\theta}(t)}(\boldsymbol{r}^*|\boldsymbol{q}) \geq C \left(1 - \pi_{\boldsymbol{\theta}(t)}(\boldsymbol{r}^*|\boldsymbol{q})\right)^2.
$$

This differential inequality yields:

$$
\pi_{\boldsymbol{\theta}(t)}(\boldsymbol{r}^*|\boldsymbol{q}) > 1 - \frac{1}{Ct + \frac{1}{1 - \pi_{\boldsymbol{\theta}(0)}(\boldsymbol{r}^*|\boldsymbol{q})}}.
$$

For any $\epsilon > 0$, there exists $T_1 = \frac{1}{C}\left(\frac{1}{\epsilon} - \frac{1}{1 - \pi_{\boldsymbol{\theta}(0)}(\boldsymbol{r}^*|\boldsymbol{q})}\right) = \mathcal{O}(\frac{1}{\epsilon})$, such that

$$
\pi_{\boldsymbol{\theta}(t)}(\boldsymbol{r}^*|\boldsymbol{q}) > 1 - \epsilon.
$$

This concludes the proof of the theorem. $\qquad\square$

### D.4 PROOF OF THEOREM 5.4

To proof Theorem 5.4, we first establish an upper bound for $\pi_{\boldsymbol{\theta}(t)}(\boldsymbol{r}'|\boldsymbol{q})$ (Eq D.6), then we analyze the ratio $\rho_{\boldsymbol{\theta}(t)}(\boldsymbol{r}_i) = \frac{\pi_{\boldsymbol{\theta}(t)}(\boldsymbol{r}_i|\boldsymbol{q})}{\pi_{\boldsymbol{\theta}(t)}(\boldsymbol{r}^*|\boldsymbol{q})}$ (Eq D.7)and prove that for $t > T_0$, $\text{ACC}_{\boldsymbol{\theta}(t)} \geq p^*(\boldsymbol{r}')$.

*Proof of Theorem 5.4.* Let $\boldsymbol{r}^*$ and $\boldsymbol{r}'$ be the optimal and second optimal reasoning patterns. For $\pi_{\boldsymbol{\theta}(t)}(\boldsymbol{r}'|\boldsymbol{q})$:

$$
\begin{aligned}
\frac{d}{dt}\pi_{\boldsymbol{\theta}(t)}(\boldsymbol{r}'|\boldsymbol{q}) &= \sum_{\boldsymbol{r}_j \in \mathcal{R}} \left(\mathrm{ACC}_{\boldsymbol{\theta}(t)} - p^*(\boldsymbol{r}_j)\right) \pi_{\boldsymbol{\theta}(t)}^2(\boldsymbol{r}_j|\boldsymbol{q})\pi_{\boldsymbol{\theta}(t)}(\boldsymbol{r}'|\boldsymbol{q}) \\
&\quad + \left(p^*(\boldsymbol{r}') - \mathrm{ACC}_{\boldsymbol{\theta}(t)}\right) \pi_{\boldsymbol{\theta}(t)}^2(\boldsymbol{r}'|\boldsymbol{q}) \\
&= \sum_{\boldsymbol{r}_j \in \mathcal{R}/\{\boldsymbol{r}^*\}} \left(\mathrm{ACC}_{\boldsymbol{\theta}(t)} - p^*(\boldsymbol{r}_j)\right) \pi_{\boldsymbol{\theta}(t)}^2(\boldsymbol{r}_j|\boldsymbol{q})\pi_{\boldsymbol{\theta}(t)}(\boldsymbol{r}'|\boldsymbol{q}) \\
&\quad + \left(p^*(\boldsymbol{r}') - \mathrm{ACC}_{\boldsymbol{\theta}(t)}\right) \pi_{\boldsymbol{\theta}(t)}^2(\boldsymbol{r}'|\boldsymbol{q}) \left(1 - \pi_{\boldsymbol{\theta}(t)}(\boldsymbol{r}^*|\boldsymbol{q})\right) \\
&\leq \sum_{\boldsymbol{r}_j \in \mathcal{R}/\{\boldsymbol{r}^*\}} \pi_{\boldsymbol{\theta}(t)}^2(\boldsymbol{r}_j|\boldsymbol{q})\pi_{\boldsymbol{\theta}(t)}(\boldsymbol{r}'|\boldsymbol{q}) \\
&\quad + \left(p^*(\boldsymbol{r}') - p^*(\boldsymbol{r}')\pi_{\boldsymbol{\theta}(t)}(\boldsymbol{r}^*|\boldsymbol{q})\right) \pi_{\boldsymbol{\theta}(t)}^2(\boldsymbol{r}'|\boldsymbol{q}) \left(1 - \pi_{\boldsymbol{\theta}(t)}(\boldsymbol{r}^*|\boldsymbol{q})\right) \\
&\leq \pi_{\boldsymbol{\theta}(t)}(\boldsymbol{r}'|\boldsymbol{q}) \left(1 - \pi_{\boldsymbol{\theta}(t)}(\boldsymbol{r}^*|\boldsymbol{q})\right)^2 + p^*(\boldsymbol{r}')\pi_{\boldsymbol{\theta}(t)}^2(\boldsymbol{r}'|\boldsymbol{q}) \left(1 - \pi_{\boldsymbol{\theta}(t)}(\boldsymbol{r}^*|\boldsymbol{q})\right)^2 \\
&\leq 2\left(1 - \pi_{\boldsymbol{\theta}(t)}(\boldsymbol{r}^*|\boldsymbol{q})\right)^2.
\end{aligned}
$$

This yields:

$$
\pi_{\boldsymbol{\theta}(t)}(\boldsymbol{r}'|\boldsymbol{q}) \leq 1 - \frac{1}{2t + \frac{1}{1-\pi_{\boldsymbol{\theta}(0)}(\boldsymbol{r}'|\boldsymbol{q})}}. \tag{D.6}
$$

Next, define $\rho_{\boldsymbol{\theta}(t)}(\boldsymbol{r}_i) = \frac{\pi_{\boldsymbol{\theta}(t)}(\boldsymbol{r}_i|\boldsymbol{q})}{\pi_{\boldsymbol{\theta}(t)}(\boldsymbol{r}^*|\boldsymbol{q})}$, then for all $\boldsymbol{r}_i \in \mathcal{R}/\{\boldsymbol{r}^*, \boldsymbol{r}'\}$, we have

$$
\begin{aligned}
\frac{d}{dt}\rho_{\boldsymbol{\theta}(t)}(\boldsymbol{r}_i) &= \frac{\pi_{\boldsymbol{\theta}(t)}(\boldsymbol{r}^*|\boldsymbol{q})\frac{d}{dt}\pi_{\boldsymbol{\theta}(t)}(\boldsymbol{r}_i|\boldsymbol{q}) - \pi_{\boldsymbol{\theta}(t)}(\boldsymbol{r}_i|\boldsymbol{q})\frac{d}{dt}\pi_{\boldsymbol{\theta}(t)}(\boldsymbol{r}^*|\boldsymbol{q})}{\pi_{\boldsymbol{\theta}(t)}^2(\boldsymbol{r}^*|\boldsymbol{q})} \\
&= \frac{\pi_{\boldsymbol{\theta}(t)}(\boldsymbol{r}_i|\boldsymbol{q})}{\pi_{\boldsymbol{\theta}(t)}(\boldsymbol{r}^*|\boldsymbol{q})} \left(\left(p^*(\boldsymbol{r}_i) - \mathrm{ACC}_{\boldsymbol{\theta}(t)}\right) \pi_{\boldsymbol{\theta}(t)}(\boldsymbol{r}_i|\boldsymbol{q}) - \left(p^*(\boldsymbol{r}^*) - \mathrm{ACC}_{\boldsymbol{\theta}(t)}\right) \pi_{\boldsymbol{\theta}(t)}(\boldsymbol{r}^*|\boldsymbol{q})\right) \\
&\leq -\rho_{\boldsymbol{\theta}(t)}(\boldsymbol{r}_i) \left(p^*(\boldsymbol{r}^*) - \mathrm{ACC}_{\boldsymbol{\theta}(t)}\right) \pi_{\boldsymbol{\theta}(t)}(\boldsymbol{r}^*|\boldsymbol{q}) \leq 0
\end{aligned}
$$

The last inequality is based on the condition for case 2 and $\mathrm{ACC}_{\boldsymbol{\theta}(t)} > \mathrm{ACC}_{\boldsymbol{\theta}(0)}$, so $p^*(\boldsymbol{r}_i) < \mathrm{ACC}_{\boldsymbol{\theta}(t)}$ holds for all $\boldsymbol{r}_i \in \mathcal{R}/\{\boldsymbol{r}^*\}, t \geq 0$. Then as $\frac{d}{dt}\rho_{\boldsymbol{\theta}(t)}(\boldsymbol{r}_i) \leq 0$, we have:

$$
\pi_{\boldsymbol{\theta}(t)}(\boldsymbol{r}^*|\boldsymbol{q}) = \frac{1 - \pi_{\boldsymbol{\theta}(t)}(\boldsymbol{r}'|\boldsymbol{q})}{\sum_{b_i \in \mathcal{R}/\{\boldsymbol{r}'\}} \rho_{\boldsymbol{\theta}(t)}(\boldsymbol{r}_i)} \geq \frac{1 - \pi_{\boldsymbol{\theta}(t)}(\boldsymbol{r}'|\boldsymbol{q})}{\sum_{b_i \in \mathcal{R}/\{\boldsymbol{r}'\}} \rho_{\boldsymbol{\theta}(0)}(\boldsymbol{r}_i)}
$$

While $\mathrm{ACC}_{\boldsymbol{\theta}(t)} < p^*(\boldsymbol{r}')$, define $\gamma_{\pi_{\mathrm{ref}}} := \sum_{\boldsymbol{r} \in \mathcal{R}/\{\boldsymbol{r}'\}} \frac{\pi_{\boldsymbol{\theta}_{\mathrm{ref}}}(\boldsymbol{r}|\boldsymbol{q})}{\pi_{\boldsymbol{\theta}_{\mathrm{ref}}}(\boldsymbol{r}^*|\boldsymbol{q})}$ we have:

$$
\begin{aligned}
\frac{d}{dt}\rho_{\boldsymbol{\theta}(t)}(\boldsymbol{r}_i) &\leq -\rho_{\boldsymbol{\theta}(t)}(\boldsymbol{r}_i) \left(p^*(\boldsymbol{r}^*) - \mathrm{ACC}_{\boldsymbol{\theta}(t)}\right) \pi_{\boldsymbol{\theta}(t)}(\boldsymbol{r}^*|\boldsymbol{q}) \\
&\leq -\rho_{\boldsymbol{\theta}(t)}(\boldsymbol{r}_i) \left(p^*(\boldsymbol{r}^*) - p^*(\boldsymbol{r}')\right) \frac{1 - \pi_{\boldsymbol{\theta}(t)}(\boldsymbol{r}'|\boldsymbol{q})}{\gamma_{\pi_{\mathrm{ref}}}} \\
&\leq -\rho_{\boldsymbol{\theta}(t)}(\boldsymbol{r}_i) \frac{p^*(\boldsymbol{r}^*) - p^*(\boldsymbol{r}')}{\gamma_{\pi_{\mathrm{ref}}}} \frac{1}{2t + \frac{1}{1-\pi_{\boldsymbol{\theta}(0)}(\boldsymbol{r}'|\boldsymbol{q})}}
\end{aligned}
$$

Then we have:

$$
\frac{d}{dt}\rho_{\boldsymbol{\theta}(t)}(\boldsymbol{r}_i) \leq -\rho_{\boldsymbol{\theta}(t)}(\boldsymbol{r}_i) \frac{p^*(\boldsymbol{r}^*) - p^*(\boldsymbol{r}')}{\gamma_{\pi_{\mathrm{ref}}}} \frac{1}{2t + \frac{1}{1-\pi_{\boldsymbol{\theta}(0)}(\boldsymbol{r}'|\boldsymbol{q})}}, \tag{D.7}
$$

this differential inequality leads to:

$$
\rho_{\boldsymbol{\theta}(t)}(\boldsymbol{r}_i) \leq \frac{\left(1 - \pi_{\boldsymbol{\theta}(0)}(\boldsymbol{r}'|\boldsymbol{q})\right)^{-\Delta/(2\gamma_{\pi_{\mathrm{ref}}})}}{\left(1/(1-\pi_{\boldsymbol{\theta}(0)}(\boldsymbol{r}'|\boldsymbol{q})) + 2t\right)^{\Delta/(2\gamma_{\pi_{\mathrm{ref}}})}} \rho_{\boldsymbol{\theta}(0)}(\boldsymbol{r}_i).
$$

Where $\Delta = p^*(\boldsymbol{r}^*) - p^*(\boldsymbol{r}')$. Recall that:

$$\mathrm{ACC}_{\boldsymbol{\theta}(t)} - p^*(\boldsymbol{r}') = \sum_{\boldsymbol{r} \in \mathcal{R}} \pi_{\boldsymbol{\theta}_{\boldsymbol{\theta}(t)}}(\boldsymbol{r}|\boldsymbol{q})p^*(\boldsymbol{r}) - p^*(\boldsymbol{r}')$$

$$= p^*(\boldsymbol{r}^*) \left[ (p^*(\boldsymbol{r}^*) - p^*(\boldsymbol{r}')) + \sum_{\boldsymbol{r}_i \in \mathcal{R}/\{\boldsymbol{r}'\}} \rho_{\boldsymbol{\theta}(t)}(\boldsymbol{r}_i) \left( p^*(\boldsymbol{r}_i) - p^*(\boldsymbol{r}') \right) \right]$$

$$\geq p^*(\boldsymbol{r}^*) \left[ (p^*(\boldsymbol{r}^*) - p^*(\boldsymbol{r}')) - \sum_{\boldsymbol{r}_i \in \mathcal{R}/\{\boldsymbol{r}'\}} \rho_{\boldsymbol{\theta}(t)}(\boldsymbol{r}_i)p^*(\boldsymbol{r}') \right].$$

Let $C_2 = 1/\Delta, C_1 = p^*(\boldsymbol{r}')/\Delta$, define:

$$T_0 = \frac{1}{2 - 2\pi_{\boldsymbol{\theta}_{\mathrm{ref}}}(\boldsymbol{r}'|\boldsymbol{q})} \left( (C_1 \cdot \gamma_{\pi_{\mathrm{ref}}})^{2C_2 \cdot \gamma_{\pi_{\mathrm{ref}}}} - 1 \right),$$

then:

$$\sum_{\boldsymbol{r}_i \in \mathcal{R}/\{\boldsymbol{r}'\}} \rho_{\boldsymbol{\theta}(T_0)}(\boldsymbol{r}_i)p^*(\boldsymbol{r}')$$

$$\leq \frac{\left( 1 - \pi_{\boldsymbol{\theta}(0)}(\boldsymbol{r}'|\boldsymbol{q}) \right)^{-\Delta/(2\gamma_{\pi_{\mathrm{ref}}})}}{\left( 1/(1 - \pi_{\boldsymbol{\theta}(0)}(\boldsymbol{r}'|\boldsymbol{q})) + 2T_0 \right)^{\Delta/(2\gamma_{\pi_{\mathrm{ref}}})}} \sum_{\boldsymbol{r}_i \in \mathcal{R}/\{\boldsymbol{r}'\}} \rho_{\boldsymbol{\theta}(0)}(\boldsymbol{r}_i)p^*(\boldsymbol{r}')$$

$$\leq \frac{\left( 1 - \pi_{\boldsymbol{\theta}(0)}(\boldsymbol{r}'|\boldsymbol{q}) \right)^{-\Delta/(2\gamma_{\pi_{\mathrm{ref}}})}}{\left( 1/(1 - \pi_{\boldsymbol{\theta}(0)}(\boldsymbol{r}'|\boldsymbol{q})) \cdot \left( \frac{\gamma_{\pi_{\mathrm{ref}}}p^*(\boldsymbol{r}')}{\Delta} \right)^{2\gamma_{\pi_{\mathrm{ref}}}/\Delta} \right)^{\Delta/(2\gamma_{\pi_{\mathrm{ref}}})}} p^*(\boldsymbol{r}') \sum_{\boldsymbol{r}_i \in \mathcal{R}/\{\boldsymbol{r}'\}} \rho_{\boldsymbol{\theta}(0)}(\boldsymbol{r}_i)$$

$$= \frac{1}{\left( \left( \frac{\gamma_{\pi_{\mathrm{ref}}}p^*(\boldsymbol{r}')}{\Delta} \right)^{2\gamma_{\pi_{\mathrm{ref}}}/\Delta} \right)^{\Delta/(2\gamma_{\pi_{\mathrm{ref}}})}} p^*(\boldsymbol{r}')\gamma_{\pi_{\mathrm{ref}}}$$

$$= \Delta = p^*(\boldsymbol{r}^*) - p^*(\boldsymbol{r}').$$

Finally, for $t > T_0$ we have:

$$\mathrm{ACC}_{\boldsymbol{\theta}(t)} - p^*(\boldsymbol{r}') \geq p^*(\boldsymbol{r}^*) \left[ (p^*(\boldsymbol{r}^*) - p^*(\boldsymbol{r}')) - \sum_{\boldsymbol{r}_i \in \mathcal{R}/\{\boldsymbol{r}'\}} \rho_{\boldsymbol{\theta}(t)}(\boldsymbol{r}_i)p^*(\boldsymbol{r}') \right]$$

$$\geq p^*(\boldsymbol{r}^*) \left[ (p^*(\boldsymbol{r}^*) - p^*(\boldsymbol{r}')) - (p^*(\boldsymbol{r}^*) - p^*(\boldsymbol{r}')) \right] = 0.$$

This concludes the proof of the theorem. $\qquad\square$

### D.5 PROOF OF THEOREM 5.6

*Proof of Theorem 5.6.* First, as we consider the case $|\mathcal{A}| = 2$ and $\pi_{\boldsymbol{\theta}_{\mathrm{base}}}(\boldsymbol{r}|q) = \frac{1}{|\mathcal{R}|}$, Assumption 5.5 holds means $\frac{1}{|\mathcal{R}|} \sum_{\boldsymbol{r} \in \mathcal{R}}(p^*(\boldsymbol{r})) > 0.5$. The derivative of overall accuracy can be written as

$$\frac{d}{dt}\mathrm{ACC}_{\boldsymbol{\theta}(t)} = \sum_{\boldsymbol{r}_i \in \mathcal{R}} p^*(\boldsymbol{r}_i) \cdot \frac{d}{dt}\pi_{\boldsymbol{\theta}(t)}(\boldsymbol{r}_i|\boldsymbol{q})$$

$$= \sum_{\boldsymbol{r}_i \in \mathcal{R}} \pi_{\boldsymbol{\theta}(t)}^2(\boldsymbol{r}_i|\boldsymbol{q}) \left( p^*(\boldsymbol{r}_i) - \mathrm{ACC}_{\boldsymbol{\theta}(t)} \right) \left( -\sum_{\boldsymbol{a} \in \mathcal{A}} \log(\pi_{\boldsymbol{\theta}(t)}(\boldsymbol{a}|\boldsymbol{r}_i, q)) - \mathrm{Con}_{\boldsymbol{\theta}(t)} \right),$$

where $\text{Con}_{\boldsymbol{\theta}(t)} = -\sum_{\boldsymbol{r}_i \in \mathcal{R}} \pi_{\boldsymbol{\theta}(t)}(\boldsymbol{r}_i|\boldsymbol{q}) \sum_{\boldsymbol{a} \in \mathcal{A}} \log(\pi_{\boldsymbol{\theta}(t)}(\boldsymbol{a}|\boldsymbol{r}_i, q))$. Here we assume $\mathcal{A} = \{\boldsymbol{a}^*, \boldsymbol{a}'\}$ where $\boldsymbol{a}'$ is the incorrect answer, and $\pi_{\boldsymbol{\theta}(0)}(\boldsymbol{r}_i|\boldsymbol{q}) = \frac{1}{|\mathcal{R}|}$ for all $\boldsymbol{r}_i \in \mathcal{R}$. Then:

$$
\begin{aligned}
\frac{d}{dt}\text{ACC}_{\boldsymbol{\theta}(t)}\Big|_{t=0} &= \sum_{\boldsymbol{r}_i \in \mathcal{R}} \frac{1}{|\mathcal{R}|^2} \left( p^*(\boldsymbol{r}_i) - \frac{1}{|\mathcal{R}|} \sum_{\boldsymbol{r}_j \in \mathcal{R}} p^*(\boldsymbol{r}_j) \right) \cdot \\
&\qquad \left( -\sum_{\boldsymbol{a} \in \mathcal{A}} \log(\pi_{\boldsymbol{\theta}(0)}(\boldsymbol{a}|\boldsymbol{r}_i, q)) + \frac{1}{|\mathcal{R}|} \sum_{\boldsymbol{r}_j \in \mathcal{R}} \sum_{\boldsymbol{a} \in \mathcal{A}} \log(\pi_{\boldsymbol{\theta}(0)}(\boldsymbol{a}|\boldsymbol{r}_j, q)) \right) \\
&= \frac{1}{|\mathcal{R}|^2} \sum_{\boldsymbol{r}_i \in \mathcal{R}} \left( p^*(\boldsymbol{r}_i) - \frac{1}{|\mathcal{R}|} \sum_{\boldsymbol{r}_j \in \mathcal{R}} p^*(\boldsymbol{r}_j) \right) \cdot \left( -\sum_{\boldsymbol{a} \in \mathcal{A}} \log(\pi_{\boldsymbol{\theta}(0)}(\boldsymbol{a}|\boldsymbol{r}_i, q)) \right).
\end{aligned}
$$

As we consider the case $|\mathcal{R}| \to +\infty$ and the success rate $p^*(\boldsymbol{r})$ are i.i.d. samples from $U[0,1]$. By the Central limit theorem, for large $|\mathcal{R}|$, the sample mean $\frac{1}{|\mathcal{R}|} \sum_{\boldsymbol{r} \in \mathcal{R}} (p^*(\boldsymbol{r}))$ is concentrated around the its expectation 0.5, the deviation is of order is $\mathcal{O}(1/\sqrt{|\mathcal{R}|})$, We define this deviation as $\delta = \mathcal{O}(1/|\mathcal{R}|^{1/2}) > 0$ such that $\frac{1}{|\mathcal{R}|} \sum_{\boldsymbol{r} \in \mathcal{R}} (p^*(\boldsymbol{r})) = 0.5 + \delta$.

According to the Gibbs conditioning principle ((Dembo, 2009) Corollary 7.3.5 and Theorem 7.3.8), as $|\mathcal{R}| \to \infty$, the empirical distribution of the $p^*(\boldsymbol{r})$ conditioned on their mean $\frac{1}{|\mathcal{R}|} \sum_{\boldsymbol{r} \in \mathcal{R}} (p^*(\boldsymbol{r})) = 0.5 + \delta$ converges to the density: $f(x) = 1 + 12\delta(x - 0.5), x \in [0,1]$. As $|\mathcal{R}| \to \infty$, we can approximate the summation over $\boldsymbol{r}$ with an integral over this limiting distribution $f(x)$. The derivative becomes:

$$
\frac{d}{dt}\text{ACC}_{\boldsymbol{\theta}(t)}\Big|_{t=0} = \int_0^1 -(1 + 12\delta(x - 0.5)) \cdot (x - 0.5 - \delta) \cdot \log(x(1-x)) \, dx = \frac{2}{3}\delta > 0
$$

Since $delta > 0$, the derivative is positive. As $|\mathcal{R}| \to \infty$, this holds with probability 1, proving the first statement.

To prove the second statement, we first analyze the RLIF reward function. For an action space of size $|\mathcal{A}| = 2$, the reward for a path $\boldsymbol{r}$ is proportional to the negative log-likelihood of the policy:

$$
R_{\text{RLIF}}(\boldsymbol{r}) - \frac{1}{|\mathcal{A}|} \sum_{\boldsymbol{a}_j \in \mathcal{A}} \log(\pi_\theta(\boldsymbol{a}_j|q, \boldsymbol{r})) = -\frac{1}{2} \log(p^*(\boldsymbol{r})(1 - p^*(\boldsymbol{r})))
$$

The function $g(p) = p(1-p)$ is maximized at $p = 0.5$. Therefore, maximizing the reward $R_{\text{RLIF}}(\boldsymbol{r})$ is equivalent to choosing the path whose success rate $p^*(\boldsymbol{r})$ is furthest from 0.5.

Let $p^*_{\min} = \min_{\boldsymbol{r} \in \mathcal{R}} p^*(\boldsymbol{r})$ and $p^*_{\max} = \max_{\boldsymbol{r} \in \mathcal{R}} p^*(\boldsymbol{r})$. The path with the highest reward will be the one with success rate $p^*_{\min}$ if it is further from 0.5 than $p^*_{\max}$ is. This condition is expressed as:

$$
0.5 - p^*_{\min} > p^*_{\max} - 0.5 \iff p^*_{\min} + p^*_{\max} < 1
$$

Thus, the event $\arg\max_{\boldsymbol{r}} R_{\text{RLIF}}(\boldsymbol{r}) = \arg\min_{\boldsymbol{r}} p^*(\boldsymbol{r})$ is equivalent to the event $p^*_{\min} + p^*_{\max} < 1$.

We now calculate the probability of this event. The success rates $p^*(\boldsymbol{r})$ are drawn from the limiting distribution $f(x) = 1 + 12\delta(x - 0.5)$. From the theory of order statistics, the scaled minimum and maximum of $n = |\mathcal{R}|$ samples converge in distribution to independent exponential random variables. Let $Y_n = n \cdot p^*_{\min}$ and $Z_n = n \cdot (1 - p^*_{\max})$. As $n \to \infty$:

- $Y_n$ converges to $Y \sim \text{Exp}(\lambda_Y)$, where the rate is the density at the lower bound: $\lambda_Y = f(0) = 1 - 6\delta$.
- $Z_n$ converges to $Z \sim \text{Exp}(\lambda_Z)$, where the rate is the density at the upper bound: $\lambda_Z = f(1) = 1 + 6\delta$.

The condition $p^*_{\min} + p^*_{\max} < 1$ can be rewritten in terms of these asymptotic variables:

$$
\frac{Y_n}{n} + \left(1 - \frac{Z_n}{n}\right) < 1 \implies Y_n < Z_n
$$

The probability of this event is $P(Y < Z)$. For two independent exponential variables, this is given by:

$$P(Y < Z) = \frac{\lambda_Y}{\lambda_Y + \lambda_Z} = \frac{1 - 6\delta}{(1 - 6\delta) + (1 + 6\delta)} = \frac{1 - 6\delta}{2} = \frac{1}{2} - 3\delta$$

Since $\delta = O(1/|\mathcal{R}|^{1/2})$, in the limit as $|\mathcal{R}| \to \infty$, $\delta \to 0$. Therefore, the limiting probability is:

$$\lim_{|\mathcal{R}| \to \infty} P(\arg\max_{\boldsymbol{r}} R_{\mathrm{RLIF}}(\boldsymbol{r}) = \arg\min_{\boldsymbol{r}} p^*(\boldsymbol{r})) = \lim_{\delta \to 0} \left( \frac{1}{2} - 3\delta \right) = \frac{1}{2}$$

This concludes the proof of the second statement.

$\square$

# E   FULL PROMPTS

## E.1   FULL PROMPT FOR REASONING PATTERNS EXTRACTION

---

**Full Prompt for Reasoning Patterns Extraction**

You will analyze multiple AI reasoning processes, showing how different models solve problems.
Analysis Steps:
For each reasoning process, identify:
Key words and recurring phrases
Logical structure of the argument
Problem-solving techniques used
Step-by-step progression
Group similar reasoning processes into exactly 5 patterns based on shared:
Vocabulary patterns (common terms and phrases)
Logical frameworks (how arguments are structured)
Solution approaches (methods used to reach conclusions)
IMPORTANT! You need to categorize general thinking patterns for problem solving, rather than categorizing types of problems themselves. In other words, each type of thinking pattern you give can be applied to any problem, rather than being task-specific. What you need to classify is the way of thinking, not the way of solving problems.
For example, you can classify them into:
1. derivation based on existing knowledge
2. derivation combined with verification
3. independent calculation, suitable for learners starting from the basics,
4. detailed manual derivation, suitable for learners who need to deeply understand each step
5. etc
Or you can classify them into:
1. knowledge call type
2. derivation and verification type
3. etc.
Output Format:
Pattern 1:
Name and description
Defining characteristics
Example processes from input
Common elements
Pattern 2:
Name and description
Defining characteristics
Example processes from input
Common elements
...
There are reasoning processes:

---

{extracted_responses}
Group similar reasoning processes into exactly 5 patterns. Please provide a classification of reasoning patterns, with each group defined by their common characteristics and examples from these processes.

## E.2 FULL PROMPT FOR RESPONSES CLASSIFICATION

**Full Prompt for Responses Classification**

Please classify each reasoning process into one of the following five patterns. Output the classifications in JSON format.
{patterns_categories}
Here are the reasoning processes to classify:
{extracted_responses}
Please output your classification in this JSON format:

## E.3 PROMPT EXAMPLES FOR EVALUATION TASKS

**Prompt Example for Easy Countdown**

<|im_start|>system
You are a helpful assistant. You first thinks about the reasoning process in the mind and then provides the user with the answer.<|im_end|>
<|im_start|>
user Using the numbers [1,2,3,7], create an equation that equals 5. You can only use basic arithmetic addition (+) and subtraction (-), and each number from the list can be used at most once. Show your work in <think> </think> tags. And return the final answer in <answer> </answer> tags, for example, to get 5 from [1,2,3,7], we have <answer> 2 + 3 = 5 </answer>; to get 13 from [1,2,3,7], we have <answer> 1 + 2 + 3 + 7 = 13 </answer>.<|im_end|>
<|im_start|>assistant
Let me solve this step by step.
<think>

**Prompt Example for Long Arithmetic**

<|im_start|>system
You are a helpful assistant. You first thinks about the reasoning process in the mind and then provides the user with the answer.<|im_end|>
<|im_start|>user
Considering the experssion (((((((((16 + 47) + 92) + 84) + 5) + 49) + 82) - 33) - 99) - 69), you need to start from the innermost bracket and work your way out. You need to give the specific calculation process. You have to do a certain amount of calculations on each line, and print each calculation in the format of a + b = c or a - b = c, for example "1 + 2 = 3".<|im_end|>
<|im_start|>assistant
Let me solve this step by step.
<think>

**Prompt Example for Number Theory**

<|im_start|>system
You are a helpful assistant.<|im_end|>
<|im_start|>user
You have four bags of copper coins. Each bag has the same number of copper coins. One day, you find a bag of 23 coins. You decide to redistribute the number of coins you have

so that all five bags you hold have the same number of coins. You successfully manage to redistribute all the coins, and you also note that you have more than 120 coins. What is the smallest number of coins you could have had before finding the bag of 23 coins?
Please reason step by step, and put your final answer within \boxed{}.<|im_end|>
<|im_start|>assistant

---

**Prompt Example for Geometry**

<|im_start|>system
You are a helpful assistant.<|im_end|>
<|im_start|>user
Two sides of scalene $\bigtriangleup ABC$ measure $4$ centimeters and $7$ centimeters. How many different whole centimeter lengths are possible for the third side?
Please reason step by step, and put your final answer within \boxed{}.<|im_end|>
<|im_start|>assistant

---

**Prompt Example for Algebra**

<|im_start|>system
You are a helpful assistant.<|im_end|>
<|im_start|>user
Six aluminum cans can be recycled to make a new can. How many new cans can eventually be made from 200 aluminum cans? (Remember that the first new cans that are made can then be recycled into even newer cans!)
Please reason step by step, and put your final answer within \boxed{}.<|im_end|>
<|im_start|>assistant

---

**Prompt Example for Calculus**

<|im_start|>system
You are a helpful assistant.<|im_end|>
<|im_start|>user
For how many values of $x$ in $[0,2\pi]$ is $\cos^{-1}(\cos 4 x) = \sin^{-1}(\sin x)$?
Please reason step by step, and put your final answer within \boxed{}.<|im_end|>
<|im_start|>assistant

---

**Prompt Example for Counting and Probability**

<|im_start|>system
You are a helpful assistant.<|im_end|>
<|im_start|>user
7 white balls and $k$ black balls are placed into a bin. Two balls are drawn at random. The probability that one ball is white and the other is black is $\frac{35}{66}$. Find the smallest possible value of $k$.
Please reason step by step, and put your final answer within \boxed{}.<|im_end|>
<|im_start|>assistant

---

**Prompt Example for AMC23-Q19**

<|im_start|>system
You are a helpful assistant.<|im_end|>
<|im_start|>user
How many digits are in the base-ten representation of '$8^5 \cdot 5^{10} \cdot 15^5$'?
Please reason step by step, and put your final answer within \boxed{}.<|im_end|>
<|im_start|>assistant

Prompt Example for AIME24-Q1

<|im_start|>system
You are a helpful assistant.<|im_end|>
<|im_start|>user
Every morning Aya goes for a $9$-kilometer-long walk and stops at a coffee shop afterwards. When she walks at a constant speed of $s$ kilometers per hour, the walk takes her 4 hours, including $t$ minutes spent in the coffee shop. When she walks $s+2$ kilometers per hour, the walk takes her 2 hours and 24 minutes, including $t$ minutes spent in the coffee shop. Suppose Aya walks at $s+\frac{1}{2}$ kilometers per hour. Find the number of minutes the walk takes her, including the $t$ minutes spent in the coffee shop.
Please reason step by step, and put your final answer within \boxed{}.<|im_end|>
<|im_start|>assistant

Prompt Example for OlympiadBench-Q1631

<|im_start|>system
You are a helpful assistant.<|im_end|>
<|im_start|>user
For a positive integer $a$, define a sequence of integers $x_{1}, x_{2}, \ldots$ by letting $x_{1}=a$ and $x_{n+1}=2 x_{n}+1$ for $n \geq 1$. Let $y_{n}=2^{x_{n}}-1$. Determine the largest possible $k$ such that, for some positive integer $a$, the numbers $y_{1}, \ldots, y_{k}$ are all prime.
Please reason step by step, and put your final answer within \boxed{}.<|im_end|>
<|im_start|>assistant

