# OpenReview forum: "Reshaping Reasoning in LLMs: A Theoretical Analysis of RL Training Dynamics through Pattern Selection"
_ICLR.cc/2026/Conference — ICLR 2026 Poster_

### Official Review · Reviewer_BMWN · 2025-10-30

**Soundness:** 2
**Presentation:** 3
**Contribution:** 2
**Rating:** 2
**Confidence:** 3

**Summary:**

This work first conducts analysis on reasoning-pattern and token levels. They claim that there are patterns obtaining stable accuracy, and distribution changes are only limited to a few tokens. Then, basing on the findings with the patterns, they theoretically analyze the training dynamics on RLVR and RLIF. In the experiment section, they conduct simulation on the RLVR and RLIF basing on their assumptions. The empirical results are consistent with theoretical claims.

**Strengths:**

1, In section 4.2, this work identifies patterns in LLM mathematical reasoning. This could provide inspire further explorations.

2, In section 5, basing on the assumptions, theoretical results are derived for RL training dynamics. Furthermore, these claims are supported by simulations in the experiment section.

**Weaknesses:**

1, My biggest concern is with the assumptions. Specifically,
- Assumption 5.1 assumes that success rate for each pattern remains the same. But this assumption is only demonstrated on one dataset; and there is not discussion on why this assumption is reasonable;
- Assumption 5.5 assumes that correct answer $a*$ has the highest probability across all possible answers, for all answers. I do not think this claim is well supported by Wang et al. (2022).
- In equation (5.3), the LLM is simplified as a one-step policy, where the distribution of $y_l$ is only determined by its previous token $y_{l-1}$. Can you explain whether this is an reasonable simplification?

2, Claims about the token distribution is not convincing; and is not connected to other parts of the manuscript. First, Table 1 is hard to interpret. Can you elaborate on how to understand this table? Second, in line 220, it claims that less than 10% of tokens experienced shift. But considering the vocabulary size and token distribution (a few tokens occupies significant probability mass), 10% rank shift could be significant. Can you also report the probability mass shift?

3, The conclusion in line 203 actually depends on what patterns are. Could you explain what these patterns are? Will the pattern for a prompt persist or change?

**Questions:**

1, Besides the simulation, could you verify your findings on a real dataset (e.g., MATH)?

---

> ### Author Response · Authors · 2025-11-23
>
> We thank the reviewer for their questions and feedback. Below, we address each concern in detail.
>
> > **W1**, My biggest concern is with the assumptions. For Assumption 5.1,5.5 and Eq 5.3:
>
> **Reply for W1**: Thank you for raising these important concerns. We appreciate the opportunity to clarify the grounding for our theoretical framework and analysis..
>
> > - Assumption 5.1 assumes that success rate for each pattern remains the same. But this assumption is only demonstrated on one dataset; and there is not discussion on why this assumption is reasonable;
>
>  **Explanation For Assumption 5.1**: To address your concern, **we provide additional experiments across various models and datasets. We kindly refer you to the last block of our response for detailed experiment results on different models and datasets**. The results demonstrate that the success rate remains stable during training across different settings, which supports our theoretical simplification. Recent works such as [1,2] also show that while the model's pass@K performance remains almost unchanged during training, RL enables the model to select CoT patterns with higher success rates for better performance; our findings align with these results.
>
> Moreover,  the main role of Assumption 5.1 in analysis is for theoretical simplification and to provide a clear bound for training dynamics. **Our theoretical framework and analysis can be extended to scenarios where reasoning pattern success rates change during training**,. Below, we explain how our theoretical analysis extends to two different cases: (1) Reasoning pattern success rates change, but the ordering of different reasoning patterns is preserved. (2) Reasoning pattern success rates change as they approach their inherent capacity limits.
>
>  - **case 1**. _Reasoning pattern success rates changes but the order for different reasoning patterns holds_, use $p^\*\_{\theta(t)}(r)$ denote the successrate for different reasoning patterns at step $t$, let $p^\*\_{\theta(t)}(r\_1) > p^\*\_{\theta(t)}(r\_2) > \cdots p^*\_{\theta(t)}(r\_n)$ holds for $t>0$, in this case, the condition for regim 1 in theorem 5.7 will holds across the training procedure, based on Eq D.5 the probabiity model choose the opeimal reasoning pattern $r_1$ will increase steality to reach $1$, resulting effective training, similarly, the condition for regim 2 in theorem 5.8 will holds until to regim 1, in this case, the probability for model to choose the suboptimal pattern remains to a relative high probability based on Eq D.6, resulting long time required to transform from regime 2 to regime 1.
> - **case 2**. reasoning pattern success rates change to reach their inherent capacity limit. Initially, RL would focus on improving the success rate of individual patterns. We can analyze this phase using the techniques from Regime 1 in Theorem 5.3. For a given reasoning pattern with a current success probability $p(t)$ and a maximum potential success rate $p^\*$, the optimization dynamics would first drive $p(t)$ towards $p^\*$. Our analysis shows the time to reach $|p^\* - p(t)| \leq \epsilon$ is $\mathcal{O}(1/\epsilon)$. Once the success rates of the primary reasoning patterns approach their respective limits (i.e., $\epsilon$ is sufficiently small), the main avenue for further reward improvement becomes shifting the probability of selecting *better* patterns. At this point, the training dynamic transitions into the scenario described by Assumption 5.1, and our main analysis holds for this crucial, later stage of training.
>
> [1] Yue, et al. Does reinforcement learning really incentivize reasoning capacity in llms beyond the base model?
>
> [2] Wen, et al. Reinforcement learning with verifiable rewards implicitly incentivizes correct reasoning in base llms

---

> ### Author Response · Authors · 2025-11-23
>
> > - **W1.2**: Assumption 5.5 assumes that correct answer  has the highest probability across all possible answers, for all answers. I do not think this claim is well supported by Wang et al. (2022).
>
> **Explanation For Assumption 5.5**:  Wang et al. (2022) [3] demonstrate that majority voting, i.e., sampling multiple potential answers and choosing the one that occurs most frequently, can effectively improve model performance. **This finding supports our assumption that the correct answer has the highest probability across all possible answers.** A straightforward explanation is that majority voting tends to choose the answer with the highest probability; the effectiveness of this method demonstrates that answers with the highest probability tend to be correct. Below, we provide a more detailed explanation:
>
> Assume we have $N$ potential answers $\{a_i\}_{i=1}^N$, where the model selects the $i$-th answer with probability $p_i$. We denote the answer with the largest probability $p^\\# = \max\{p_1,\cdots,p_N\}$ as $a^\\#$. Recall that majority voting first samples $K$ candidates from the base model and selects the answer with the largest occurrence as the final answer $\hat{a}$. Then, based on Theorem 1 in [4], we have that the probability that the final answer $\hat{a}$ is the answer with the largest probability $a^\\#$ satisfies:
> $$P(\hat{a} = a^\\#) = 1 - \mathcal{O}(c^{-K})$$
> where $c$ is a constant larger than 1. In other words, majority voting will converge to selecting the answer with the largest probability from the base model. Given the fact that majority voting has been shown to improve model performance, the answer with the largest probability is likely to be the correct answer $a^*$; thus, Assumption 5.5 tends to hold for many questions.
>
> We apologize for any misunderstanding or confusion caused by our statement and explanation regarding Assumption 5.5. We have added detailed explanations and revised the relevant sections accordingly in the updated manuscript.
>
> [3] Wang, et al. Self-consistency improves chain of thought reasoning in language models
>
> [4] Wu, et al. Inference scaling laws: An empirical analysis of compute-optimal inference for problem-solving with language models
>
> > - **W1.3** : Can you explain whether this is an reasonable simplification?
>
> **Explanation For Eq 5.3**, LLM is simplified as a one-step policy for training dynamic analysis: This is a good question! Due to the complexity of practical models, establishing optimization guarantees for understanding RL training dynamics has proven very challenging [5,6]. Therefore, in this paper, we consider a tabular policy parameterization, which is widely accepted for understanding RL training dynamics [7,8]. **Moreover, based on our token ranking shift experiments, such a simplification is reasonable for grasping the key dynamics in RL training**: we only need to focus on the key positions, where tokens at these positions dominate the reasoning pattern selection process. Below is a detailed explanation:
>
> Based on our ranking shift experiments, given a response $[c\_1,c\_2,\cdots,c\_n]$ sampled from the base model, only tokens on a small portion of positions experience ranking shifts after RL. That is, for most positions, the rank of the probability $\pi(c_i|c_1,\cdots,c_{i-1})$ of the sampled token $c_i$ given context $[c_1,\cdots,c_{i-1}]$ among the vocabulary $\pi(\cdot|c_1,\cdots,c_{i-1})$ remains unchanged during training. If we assume only one position exhibits a token ranking shift, denoted as $c_k^*$, then the reasoning pattern selection process can be simplified using a one-step policy, as we only need to focus on the probability $\pi(\cdot | c_1,\cdots,c_{k-1})$ to select different reasoning patterns. In this case, using Equation (5.3) to formulate the RL reasoning process is a reasonable choice for understanding the RL training process.
>
> In conclusion, the one-step policy simplification is widely accepted for theoretical RL training analysis. Moreover, based on our experiments, the reasoning pattern selection process during RL training arises from a small subset of positions. Focusing on the key positions that dominate the reasoning pattern selection process is a reasonable simplification to help us understand the RL training dynamics for LLMs.
>
> [5] Agarwal, et al. On the theory of policy gradient methods: Optimality, approximation, and distribution shift
>
> [6] Li, et al. Softmax policy gradient methods can take exponential time to converge
>
> [7] Cui, et al. "The entropy mechanism of reinforcement learning for reasoning language models
>
>
>
> [8] Zhang, Yanzhi, et al. "No Free Lunch: Rethinking Internal Feedback for LLM Reasoning." arXiv preprint arXiv:2506.17219 (2025).

---

> ### Author Response · Authors · 2025-11-23
>
> > **W2**:  Claims about the token distribution is not convincing. Can you elaborate on how to understand table1? Second, in line 220, it claims that less than 10% of tokens experienced shift. But considering the vocabulary size and token distribution, 10% rank shift could be significant.
>
> **Reply for W2**: We apologize for any misunderstanding caused. We appreciate the chance to clarify our experimental setting: **we are NOT analyzing 10% of the vocabulary tokens.** Instead, **for each response generated by the base model, only ~10% of the token _positions_ in that response show rank changes after RL tuning.** This means we can focus on these critical positions to understand how RL alters reasoning patterns. Below, we respectfully provide a step-by-step breakdown of our token-level analysis:
>
> 1. Step 1, Generate a response from base model: we first sample a response $C = [c\_1,c\_2,\dots,c\_n]$ from the base model $\pi$ given prompt $p$. For example, this might be a 50-token solution to a gsm8k problem.
> 2. Step 2, Examine each token position: For each position i in the response (i = 1 to n):
> 		- We compute the probability distribution over the entire vocabulary in both models, given the preceding context $[c_1, ..., c_{i-1}]$
> 		- We determine the rank of token $c_i$, the rank of the probability that model choose $c_i$ compared with other tokens in vacabulary, resulting the $c_i$'s rank in the base model: $r(c_i| [c_1, ..., c_{i-1}])$ and the the RL model: $r'(c_i| c_1, ..., c_{i-1})$.
> 3. Step 3, Identify positions with rank shifts: we mark a token at position $i$ as exhibiting a "rank shift" if $r(c_i| [c_1, ..., c_{i-1}]) ≠ r'(c_i| [c_1, ..., c_{i-1}])$. This indicates that RL training has changed the relative token preference at this decision point.
>
>  Based on our experiments, we find that across a response of length n, fewer than 10% of the positions exhibit rank shifts. For instance, in a 50-token response, typically only tokens in less than ~5 positions show changes. This reveals the key mechanism by which RL training alters the reasoning pattern distribution, making it reasonable to only focus on some key positions when theoretically analyzing the RL training dynamics. We have made our token-level analysis clearer in our revised version.
>
> > **W3**, The conclusion in line 203 actually depends on what patterns are. Could you explain what these patterns are? Will the pattern for a prompt persist or change?
>
> **Reply for W3**: In our reasoning pattern-level analysis, **"reasoning patterns" are classified based on keywords and logical structures**. We first collect responses from the base model and employ GPT-4o to group them into distinct categories based on keywords and logical structures, also generating a description for each pattern. We then use these classifications to analyze the reasoning pattern distribution and their corresponding success rates throughout the RL training procedure. In Table 2 in the appendix, we provide examples of the reasoning patterns we extracted and their associated keywords.
>
> As decoding is a stochastic procedure, for a given question, the model may adopt different problem-solving strategies to solve the problem; thus, the pattern can change. Moreover, during Rl training, for the same prompt, the reasoning pattern distribution will change during training, which is the main mechanism how RL affect model's performance.
>
> > **Q1**: Besides the simulation, could you verify your findings on a real dataset (e.g., MATH)?
>
> **R1**: Thank you for pointing this out. In our simulation, we show that depending on the base model, RL training dynamics can be very different. One case may be that RL is very efficient, steadily optimizing towards the optimal reasoning pattern; another case may be very difficult to optimize. Interestingly, we observe such phenomena in Qwen2.5-3B and Llama3.2-Instruct trained on the GSM8K dataset:
>
> **For Qwen2.5-3B (the model we use in our paper):** We observe the model optimizing towards the reasoning pattern with the highest success rate during training, aligning with the case we discussed for Regime 1.
>
> | pattern| base| step20| step40|step60|step80|step100|
> |-|-|-|-|-|-|-|
> | ratio of optimal patter| 0.438|0.535|0.596|0.590|0.604| 0.633|
>
> **For Llama3.2-Instruct:** We observe that the ratio of optimal reasoning patterns does not steadily increase, aligning with the case we discussed for Regime 2.
>
> |pattern|base| step20| step40|step60|step80|step100|
> |-|-|-|-|-|-|-|
> | ratio of optimal pattern|0.162|0.31|0.265|0.233|0.308|0.338|
>
> Moreover, recent work shows that while RLVR can succeed in Qwen-series models, it often struggles when optimizing Llama-series models[9,10], which aligns well with our experimental results.
>
> [9] Gandhi, et al. Cognitive behaviors that enable self-improving reasoners, or, four habits of highly effective stars
>
> [10] Zeng, et al. Simplerl-zoo: Investigating and taming zero reinforcement learning for open base models in the wild

---

> ### Author Response · Authors · 2025-11-23
>
> **Additional Experiments: Reasoning Pattern Accuracy Across RLVR/RLIF Trained on Different Base Models and Datasets**
>
> We conducted RLIF and RLVR training across different base models (Qwen, LLaMA) and training datasets (MATH, GSM8K, CodeContest). We sampled questions from the MATH dataset to check the corresponding pattern accuracy. Since different base models exhibit different reasoning patterns, we selected reasoning patterns with the top-3 frequencies and report their accuracy throughout training.
>
> **LLama3.2-3B-Instruct as base model**:
> >
> > _**MATH-RLIF**_
> >
> >| pattern-acc    | base   | step20 | step40 | step60 | step80 | step100 |
> >|--------------|--------|--------|--------|--------|--------|---------|
> >| pattern1 acc | 0.505  | 0.497  | 0.514  | 0.499  | 0.506  | 0.505   |
> >| pattern2 acc | 0.094  | 0.085  | 0.085  | 0.066  | 0.085  | 0.057   |
> >| pattern3 acc | 0.080  | 0.089  | 0.103  | 0.103  | 0.085  | 0.094   |
> >
> > _**MATH-RLVR**_
> >
> >| patterns-acc     | base   | step20 | step40 | step60 | step80 | step100 |
> >|--------------|--------|--------|--------|--------|--------|---------|
> >| pattern1 acc | 0.505  | 0.497  | 0.514  | 0.499  | 0.506  | 0.505   |
> >| pattern2 acc | 0.094  | 0.085  | 0.085  | 0.085  | 0.094  | 0.094   |
> >| pattern3 acc | 0.080  | 0.113  | 0.089  | 0.080  | 0.141  | 0.117   |
> >
> >_**GSM8K-RLIF**_
> >
> >| patterns     | base   | step20 | step40 | step60 | step80 | step100 |
> >|--------------|--------|--------|--------|--------|--------|---------|
> >| pattern1 acc | 0.505  | 0.5135 | 0.539  | 0.5135 | 0.5177 | 0.495   |
> >| pattern2 acc | 0.094  | 0.085  | 0.123  | 0.047  | 0.076  | 0.085   |
> >| pattern3 acc | 0.080  | 0.080  | 0.127  | 0.094  | 0.084  | 0.136   |
> >
> >_**GSM8K-RLVR**_
> >
> >| patterns     | base   | step20 | step40 | step60 | step80 | step100     |
> >|--------------|--------|--------|--------|--------|--------|-------------|
> >| pattern1 acc | 0.505  | 0.461  | 0.484 | 0.526 | 0.516 | 0.511      |
> >| pattern2 acc | 0.094 | 0.1033 | 0.093 | 0.12   | 0.1333 | 0.113      |
> >| pattern3 acc | 0.080  | 0.094  | 0.070  | 0.080  | 0.075  | 0.103 |
>
> **Qwen2.5-3B as base model**:
> >
> > _**GSM8K-RLIF**_
> >
> > | patterns     | base   | step20 | step40 | step60 | step80 | step100 |
> > |--------------|--------|--------|--------|--------|--------|---------|
> > | pattern1 acc | 0.3355 | 0.3617 | 0.3377 | 0.366  | 0.3725 | 0.39    |
> > | pattern2 acc | 0.294  | 0.329  | 0.321  | 0.334  | 0.334  | 0.358   |
> > | pattern3 acc | 0.2021 | 0.2074 | 0.1915 | 0.234  | 0.234  | 0.2128  |
> >
> > _**GSM8K-RLVR**_
> >
> > | patterns     | base   | step20 | step40 | step60 | step80 | step100 |
> > |--------------|--------|--------|--------|--------|--------|---------|
> > | pattern1 acc | 0.3355 | 0.3529 | 0.3747 | 0.3834 | 0.3834 | 0.4139  |
> > | pattern2 acc | 0.294  | 0.332  | 0.350  | 0.385  | 0.347  | 0.369   |
> > | pattern3 acc | 0.2021 | 0.234  | 0.2713 | 0.2074 | 0.2181 | 0.2074  |
> >
> > _**Codecontest-RLIF**_, For this experiment, we use 50 code tasks sampled from HumanEval for evaluation, including training process performance analysis and reasoning pattern analysis.
> >
> > | patterns     | base   | step20 | step40 | step60 | step80 | step100 |
> > |--------------|--------|--------|--------|--------|--------|---------|
> > | pattern1 acc | 0.398  | 0.393  | 0.401  | 0.401  | 0.401  | 0.390   |
> > | pattern2 acc | 0.108  | 0.099  | 0.099  | 0.162  | 0.342  | 0.180   |
> > | pattern3 acc | 0.756  | 0.779  | 0.814  | 0.826  | 0.727  | 0.721   |
>
> We observe that reasoning pattern accuracy remains stable during training, further validating our experimental observations and supporting Assumption 5.1.

---

> > ### Comment · Reviewer_BMWN · 2025-11-26
> > **Informative responses**
> >
> > I first thank the authors for providing such informative responses, especially empirical results to demonstrate: 1, assumption 5.1 is reasonable; 2, pattern distributions is aligned with theoretical analysis. My concerns are mostly addressed. In general, this work theoretically explains how reasoning patterns change during RL training; their findings and results are consistent with current findings. I believe this is a good contribution to the community. So, I decide to raise my score.
> >
> > On the other hand, I still have questions about assumption 5.5. For my understanding, this assumption assumes that for all prompt $x$, its correct response will be assigned the highest probability. That said, if I used beam sampling with beam number 1, I could obtain almost 100% accuracy rate. Is my understanding correct? If so, I do NOT think this is the case with most LLMs; nor [1] could support this. If my understanding is wrong, I am happy to learn your interpretation on assumption 5.5.
> >
> > [1] Wu, et al. Inference scaling laws: An empirical analysis of compute-optimal inference for problem-solving with language models

---

> > > ### Author Response · Authors · 2025-11-27
> > >
> > > We are deeply grateful for feedback and your decision to raise the score. We are happy to provide further clarification regarding Assumption 5.5.
> > >
> > > > I still have questions about assumption 5.5. For my understanding, this assumption assumes that for all prompt , its correct response will be assigned the highest probability. That said, if I used beam sampling with beam number 1, I could obtain almost 100% accuracy rate. Is my understanding correct?
> > >
> > > This is an excellent question. In fact Assumption 5.5 does not guarantee that beam search (with beam size 1 or larger) will achieve high accuracy. The key distinction lies in the difference between the **marginal probability** of an answer (which Assumption 5.5 considers) and the **joint probability** of a specific reasoning path (which Beam Search optimizes).
> > >
> > > 1.  **Beam Search:** Seeks the single sequence of tokens (reasoning $r$ and answer $a$) that maximizes the **joint probability**:
> > >     $$ \max_{r, a} \pi(r, a | x) = \max_{r, a} \pi(r|x)\pi(a|x, r) $$
> > >
> > > 2.  **Assumption 5.5:** Assume that the correct answer $a^\*$ has the highest **marginal probability** when summed over *all* possible reasoning paths $\mathcal{R}$:
> > >     $$ p(a^\*|x) = \sum_{r \in \mathcal{R}} \pi(r|x)\pi(a^\*|x, r) > p(a'|x) \quad \forall a' \neq a^\* $$
> > >
> > > **It is entirely possible that Assumption 5.5 holds but Beam Search fails.** This happens when the correct answer $a^*$ has the highest *total* probability, but a specific incorrect reasoning path has the highest *individual* likelihood.
> > >
> > > **Counter-Example:**
> > > Consider a prompt $x$ with two reasoning patterns ($r_1, r_2$) and two answers ($a^\*$ is correct, $a'$ is incorrect).
> > > *   **Reasoning Distribution:** $\pi(r_1|x) = 0.7$ (Model prefers $r_1$), $\pi(r_2|x) = 0.3$.
> > > *   **Answer Distribution:**
> > >     *   Under $r_1$: $\pi(a^*|x, r_1) = 0.4$, $\pi(a'|x, r_1) = 0.6$.
> > >     *   Under $r_2$: $\pi(a^*|x, r_2) = 0.9$, $\pi(a'|x, r_2) = 0.1$.
> > >
> > > In this scenario:
> > >
> > > **1. Assumption 5.5 Holds (Marginal Probability):**
> > > The total probability of the correct answer $a^\*$ is:
> > > $$ p(a^*|x) = (0.7 \times 0.4) + (0.3 \times 0.9) = 0.28 + 0.27 = \mathbf{0.55} $$
> > > The total probability of the incorrect answer $a'$ is:
> > > $$ p(a'|x) = (0.7 \times 0.6) + (0.3 \times 0.1) = 0.42 + 0.03 = 0.45 $$
> > > Since $0.55 > 0.45$, the assumption holds.
> > >
> > > **2. Beam Search Fails (Joint Probability):**
> > > - For beam size 1, the beam search tend to first choose $r_1$ as it have larger probability then $r_2$ and then choose the incorrect answer $a'$.
> > > - For larger beam size, beam search looks for the path with the highest joint probability. Let us calculate the probability for each path:
> > > 	*   Path 1 ($r_1 \rightarrow a'$): $0.7 \times 0.6 = \mathbf{0.42}$
> > > 	*   Path 2 ($r_1 \rightarrow a^*$): $0.7 \times 0.4 = 0.28$
> > > 	*   Path 3 ($r_2 \rightarrow a^*$): $0.3 \times 0.9 = 0.27$
> > > 	*   Path 4 ($r_2 \rightarrow a'$): $0.3 \times 0.1 = 0.03$
> > >
> > > 	In this case, beam search tend to select **Path 1** because it has the highest single-path probability ($0.42$), resulting in the **incorrect answer** $a'$.
> > >
> > > In conclusion, Assumption 5.5 does not imply Beam Search will be 100% accurate.  Mreover, Majority Voting  approximates the marginal probability $p(a|x)$ by sampling multiple paths,  choose the answer with the highest probability; the effectiveness of this method demonstrates that answers with the highest probability tend to be correct for many tasks, which support our assumption.
> > >
> > > We hope this clarifies the relationship between our assumption and beam search. We are happy to answer any further questions.

---

### Official Review · Reviewer_MU4Q · 2025-11-01

**Soundness:** 3
**Presentation:** 4
**Contribution:** 3
**Rating:** 6
**Confidence:** 3

**Summary:**

The paper analyzes how RL post-training changes LLM reasoning, showing empirically that RLVR promotes higher-success reasoning patterns while RLIF remains unstable. A two-stage q → r → a model and tabular gradient-flow analysis explain these behaviors, and controlled experiments support the overall conclusions.

**Strengths:**

The paper effectively combines empirical findings with a clean theoretical analysis, offering useful insights for understanding LLM reasoning behavior.

**Weaknesses:**

The experimental scope is narrow: all evaluations are conducted on math reasoning tasks and within the Qwen2.5 model family. Prior work has shown RLIF to produce gains on non-math general domains, so broader experiments across model families and tasks are necessary to substantiate the generality of the conclusions.

**Questions:**

1. How is pattern-level accuracy evaluated? A single response may contain multiple reasoning patterns—how do you attribute correctness to individual patterns in such mixed instances?
2. How should the proposed training-dynamics framework inform practical RLVR training?

---

> ### Author Response · Authors · 2025-11-23
>
> We sincerely appreciate the time and effort you spent on thoughtful reviews and comments. We address your comments below:
>
> > **W1**: The experimental scope is narrow. Prior work has shown RLIF to produce gains on non-math general domains, so broader experiments across model families and tasks are necessary to substantiate the generality of the conclusions.
>
> **Reply for W1**: Thank you for the suggestion! **We have added additional experiments on various models (Qwen/Llama) and datasets (MATH/CODE/GSM8K) to address your concern**. Specifically, we conducted RLIF training on the CodeContest dataset, and test it on a subset that consists 50 questions from humaneval. We find that RLIF can provide performance gains at early training steps, but the performance decreases with continued training.
>
> |models | base| Step20|Step40|Step60|Step80|Step100|
> |-|-|-|-|-|-|-|
> |pass rate|0.291|0.353|0.343|0.325|0.286|0.242|
>
>  This finding aligns with our experiments on math datasets. In our theoretical analysis section, we further provide an explanation for the early training gains, as well as the potential performance degradation of RLIF. **We kindly refer you to the last block of our response, where we conduct RLIF/RLVR on different datasets and base models**
>
> Beyond experiments, we also want to highlight our theoretical contribution: while previous work about analysis of RL for LLMs always remains empirically [1,2] or failed to prvoide a detialed analysis for the whole RL training process[3,4], we build a theoretical framework to analyze RL training dynamics for LLMs, and provide theoretical explanation for experimental observations for both RLIF and RLVR. We kindly refer you to our answer to Q2 for a more detailed explanation of our theoretical contributions and their connections to RL training.
>
> [1] Yue, et al. Does reinforcement learning really incentivize reasoning capacity in llms beyond the base model?
>
> [2] Huan, et al. Does Math Reasoning Improve General LLM Capabilities? Understanding Transferability of LLM Reasoning
>
> [3] Zhang, et al. No Free Lunch: Rethinking Internal Feedback for LLM Reasoning
>
> [4] Cui et al. The entropy mechanism of reinforcement learning for reasoning language models
>
> > **Q1**. How is pattern-level accuracy evaluated? A single response may contain multiple reasoning patterns—how do you attribute correctness to individual patterns in such mixed instances?
>
> **R1**: Thank you for this important question. Using GPT-4o, we first extract reasoning patterns from sampled model responses, then classify responses from different training checkpoints into these patterns. Responses that can't be confidently classified are labeled "Other". **Our experiments show that such ambiguous cases are rare**. The top three patterns, shown in Figure 1(b), account for 98% of responses.
>
> Once we get the pattern for each response from the base model, we can compute the reasoning pattern accuracy. As described in Appendix C.3, we estimate pattern-level accuracy by first constructing an evaluation set consisting of prompts, where each prompt consists of a question and less than half of the base model's response reasoning process and ensuring it excludes final answer) to guide the model through a specific reasoning pattern.
>
> > **Q2**. How should the proposed training-dynamics framework inform practical RLVR training?
>
> **R2** Thank you for this question! We appreciate the opportunity to clarify the practical implications of our theoretical analysis for RLVR training.
>
> Theorems 5.3 and 5.4 demonstrate that RLVR's effectiveness is highly dependent on the base model's initialization, as it can either train effectively or get stuck at suboptimal points. This theoretical finding explains widely observed practical phenomena like the success of RL for LLMs is heavily dependent on the quality of the initial base model [5,6], and can provide further insights into efficient RL training, such as incorporating supervised finetuning before RL as a cold start [7].
>
> Besides, our theoretical framework may be applied to design efficient training algorithms, such as reweighting the reward for different reasoning patterns based on the model's current pattern distribution for more efficient training. As this may be somewhat beyond the scope of this paper, we plan to leave this for future investigation.
>
> Moreover, our theoretical analysis of RLIF explains the initial performance gain of RLIF [3,8], and explains why the model's performance may decrease with continued training, which aligns with observations in [8].
>
> [5] Zeng, et al. Simplerl-zoo: Investigating and taming zero reinforcement learning for open base models in the wild
>
> [6] Gandhi, et al. Cognitive behaviors that enable self-improving reasoners, or, four habits of highly effective stars
>
> [7] Guo, et al. Deepseek-r1: Incentivizing reasoning capability in llms via reinforcement learning
>
> [8] Zhao, et al. Learning to reason without external rewards arXiv

---

> ### Author Response · Authors · 2025-11-23
>
> **Additional Experiments: Reasoning Pattern Accuracy Across RLVR/RLIF Trained on Different Base Models and Datasets**
>
> We conducted RLIF and RLVR training across different base models (Qwen, LLaMA) and training datasets (MATH, GSM8K, CodeContest). We sampled questions from the MATH dataset to check the corresponding pattern accuracy. Since different base models exhibit different reasoning patterns, we selected reasoning patterns with the top-3 frequencies and report their accuracy throughout training.
>
> **LLama3.2-3B-Instruct as base model**:
> >
> > _**MATH-RLIF**_
> >
> >| pattern-acc    | base   | step20 | step40 | step60 | step80 | step100 |
> >|--------------|--------|--------|--------|--------|--------|---------|
> >| pattern1 acc | 0.505  | 0.497  | 0.514  | 0.499  | 0.506  | 0.505   |
> >| pattern2 acc | 0.094  | 0.085  | 0.085  | 0.066  | 0.085  | 0.057   |
> >| pattern3 acc | 0.080  | 0.089  | 0.103  | 0.103  | 0.085  | 0.094   |
> >
> > _**MATH-RLVR**_
> >
> >| patterns-acc     | base   | step20 | step40 | step60 | step80 | step100 |
> >|--------------|--------|--------|--------|--------|--------|---------|
> >| pattern1 acc | 0.505  | 0.497  | 0.514  | 0.499  | 0.506  | 0.505   |
> >| pattern2 acc | 0.094  | 0.085  | 0.085  | 0.085  | 0.094  | 0.094   |
> >| pattern3 acc | 0.080  | 0.113  | 0.089  | 0.080  | 0.141  | 0.117   |
> >
> >_**GSM8K-RLIF**_
> >
> >| patterns     | base   | step20 | step40 | step60 | step80 | step100 |
> >|--------------|--------|--------|--------|--------|--------|---------|
> >| pattern1 acc | 0.505  | 0.5135 | 0.539  | 0.5135 | 0.5177 | 0.495   |
> >| pattern2 acc | 0.094  | 0.085  | 0.123  | 0.047  | 0.076  | 0.085   |
> >| pattern3 acc | 0.080  | 0.080  | 0.127  | 0.094  | 0.084  | 0.136   |
> >
> >_**GSM8K-RLVR**_
> >
> >| patterns     | base   | step20 | step40 | step60 | step80 | step100     |
> >|--------------|--------|--------|--------|--------|--------|-------------|
> >| pattern1 acc | 0.505  | 0.461  | 0.484 | 0.526 | 0.516 | 0.511      |
> >| pattern2 acc | 0.094 | 0.1033 | 0.093 | 0.12   | 0.1333 | 0.113      |
> >| pattern3 acc | 0.080  | 0.094  | 0.070  | 0.080  | 0.075  | 0.103 |
>
> **Qwen2.5-3B as base model**:
> >
> > _**GSM8K-RLIF**_
> >
> > | patterns     | base   | step20 | step40 | step60 | step80 | step100 |
> > |--------------|--------|--------|--------|--------|--------|---------|
> > | pattern1 acc | 0.3355 | 0.3617 | 0.3377 | 0.366  | 0.3725 | 0.39    |
> > | pattern2 acc | 0.294  | 0.329  | 0.321  | 0.334  | 0.334  | 0.358   |
> > | pattern3 acc | 0.2021 | 0.2074 | 0.1915 | 0.234  | 0.234  | 0.2128  |
> >
> > _**GSM8K-RLVR**_
> >
> > | patterns     | base   | step20 | step40 | step60 | step80 | step100 |
> > |--------------|--------|--------|--------|--------|--------|---------|
> > | pattern1 acc | 0.3355 | 0.3529 | 0.3747 | 0.3834 | 0.3834 | 0.4139  |
> > | pattern2 acc | 0.294  | 0.332  | 0.350  | 0.385  | 0.347  | 0.369   |
> > | pattern3 acc | 0.2021 | 0.234  | 0.2713 | 0.2074 | 0.2181 | 0.2074  |
> >
> > _**Codecontest-RLIF**_, For this experiment, we use 50 code tasks sampled from HumanEval for evaluation, including training process performance analysis and reasoning pattern analysis.
> >
> > | patterns     | base   | step20 | step40 | step60 | step80 | step100 |
> > |--------------|--------|--------|--------|--------|--------|---------|
> > | pattern1 acc | 0.398  | 0.393  | 0.401  | 0.401  | 0.401  | 0.390   |
> > | pattern2 acc | 0.108  | 0.099  | 0.099  | 0.162  | 0.342  | 0.180   |
> > | pattern3 acc | 0.756  | 0.779  | 0.814  | 0.826  | 0.727  | 0.721   |
>
> We observe that reasoning pattern accuracy remains stable during training, further validating our experimental observations and supporting Assumption 5.1.

---

### Official Review · Reviewer_xtTG · 2025-11-01

**Soundness:** 3
**Presentation:** 3
**Contribution:** 3
**Rating:** 6
**Confidence:** 2

**Summary:**

This paper provides new insights into the reinforcement learning (RL) training dynamics of large language models (LLMs) through a combination of empirical analysis and theoretical modeling. The authors focus on two reward types: verifiable rewards (RLVR) and internal feedback (RLIF). They find that RL primarily optimizes a sparse subset of “critical” tokens, which reshapes the distribution of reasoning patterns. For RLVR the authors show two scenarios, one were optimization converges efficiently and another where weaker base models experience slow convergence. For RLIF the authors show that it is initally improving performance but can degrade performance with continued training.

**Strengths:**

- Provides an explanation for the stability of RLVR and the instability of RLIF
- Relevant topic: Understanding the behavior of RL training in the context of LLMs
- Multiple datasets and model scales are used which strengthens the generality of the findings.
- Introduces an interesting and interpretable analysis framework at both the token and reasoning-pattern levels

**Weaknesses:**

- The paper would benefit from a more comprehensive description of the experimental setup for the different training/evaluation runs. Important implementation details (e.g. temperature settings, sampling strategies, and other hyper parameters) are missing making it difficult to fully reproduce or interpret the reported results.
- It remains unclear which exact implementation was used for RLVR training. For example, if vanilla GRPO was used, information about rollout numbers, optimization parameters, and possible comparisons to more recent methods (e.g., CISPO[1] or Dr.GRPO[2]) would help contextualize the empirical findings and assess their robustness.

1: MiniMax-M1: Scaling Test-Time Compute Efficiently with Lightning Attention
2: Understanding R1-Zero-Like Training: A Critical Perspective

**Questions:**

- **Q1:** For generating base model responses in the token-level analysis, was the sampling temperature set to 0?
- **Q2:** Do the authors expect qualitatively different dynamics for different RLVR variants (e.g., GRPO, PPO, or CISPO), or would the theoretical conclusions remain consistent across these variants?
- **Q3:** Could the authors elaborate on Assumption 5.1? In practice, the success rate of individual reasoning patterns may change during training. How realistic is it to assume this remains constant, and how might deviations affect the theoretical results?
- **Q4:** How sensitive are the observed “sparse token optimization” effects to the choice of base model or training dataset?

---

> ### Author Response · Authors · 2025-11-23
>
> We sincerely appreciate the time and effort you spent on thoughtful reviews and comments. We address your comments below:
>
> > **W1**: The paper would benefit from a more comprehensive description of the experimental setup for the different training/evaluation runs.
>
> **Reply for W1**: Thanks for your suggestion! We have added all the details for the experimental setup in our appendix. Specifically, for the training procedure experiments for RLVR and RLIF (Section 4.1), all our settings are exactly the same as those in the Verl framework using the GRPO algorithm. The temperature is set to 1 during training. The learning rate is set to 1e-6. The train_batch_size is set to 1024 while the ppo_mini_batch_size is set to 256. In group sampling, the number of rollouts for a single prompt is set to 5. We disable top-k and set top-p to 1.
>
> For the reasoning-pattern level (Section 4.2) and token-level (Section 4.3) experiments, we use the same training settings as described above. During inference, we set the sampling temperatures to 1.0 and 0.4, respectively, to ensure a certain level of diversity. We subsequently supplement the experiments with different temperatures (including 0 and 1).
>
> > **W2**: It remains unclear which exact implementation was used for RLVR training.
>
> **Reply for W2**: We apologize for any confusion. We used vanilla GRPO as our training algorithm for RLVR and RLIF in our experiments. For theoretical analysis, we consider the idealized policy gradient for RL training. While our analysis considers the widely adopted idealized policy gradient descent with infinite data [1,2], it captures the core objective of RL training and can provide insights that are broadly applicable to these variants. We kindly refer you to our answer to Q2 for a more detailed explanation.
>
> [1] Razin, Noam, et al. "What makes a reward model a good teacher? an optimization perspective." _arXiv preprint arXiv:2503.15477_ (2025).
>
> [2] Cui, Ganqu, et al. "The entropy mechanism of reinforcement learning for reasoning language models." _arXiv preprint arXiv:2505.22617_ (2025).
>
> > **Q1**: For generating base model responses in the token-level analysis, was the sampling temperature set to 0?
>
>
>
> _R1_: Thank you for this insightful question. We have added detailed experimental settings in our revised version. For our token-level analysis, we used a temperature of 0.4 when sampling responses from the base model. This allows us to compare token rankings between the base model and RL-tuned models in a relatively more diverse setting. Since we don't use greedy decoding, the corresponding token ranking in the base model isn't necessarily 1. We appreciate your suggestion to use temperature 0 and have conducted token ranking shift experiments with different temperatures. The results are as follows:
>
> Token ranking shift at temperature = 0, (with qwen2.5-3B trained MATH dataset)
>
> | Task   | Method | Step20 | Step40 | Step60 | Step80 | Step100 |
> |--------|--------|--------|--------|--------|--------|---------|
> | GSM8K  | RLVR   | 0.048  | 0.055  | 0.061  | 0.066  | 0.068   |
> |        | RLIF   | 0.064  | 0.074  | 0.081  | 0.087  | 0.093   |
> | MATH   | RLVR   | 0.049  | 0.053  | 0.056  | 0.059  | 0.061   |
> |        | RLIF   | 0.056  | 0.064  | 0.071  | 0.077  | 0.081   |
> | AIME24 | RLVR   | 0.045  | 0.049  | 0.052  | 0.054  | 0.056   |
> |        | RLIF   | 0.046  | 0.055  | 0.063  | 0.070  | 0.075   |
>
> Token ranking shift at temperature = 1, (with qwen2.5-3B trained MATH dataset)
>
> | Task   | Method | Step20 | Step40 | Step60 | Step80 | Step100 |
> |--------|--------|--------|--------|--------|--------|---------|
> | GSM8K  | RLVR   | 0.095  | 0.103  | 0.110  | 0.115  | 0.117   |
> |        | RLIF   | 0.108  | 0.123  | 0.130  | 0.137  | 0.144   |
> | MATH   | RLVR   | 0.093  | 0.102  | 0.108  | 0.112  | 0.115   |
> |        | RLIF   | 0.103  | 0.115  | 0.124  | 0.130  | 0.136   |
> | AIME24 | RLVR   | 0.107  | 0.117  | 0.128  | 0.135  | 0.140   |
> |        | RLIF   | 0.122  | 0.137  | 0.144  | 0.159  | 0.165   |
>
> We observe that while higher temperatures introduce more noise and result in higher ranking shift ratios, even in a high noise setting(temperature = 1), the ratio remains at a low level. This validates our finding that RL optimizes only a small subset of key tokens to reshape the reasoning pattern selection procedure.

---

> ### Author Response · Authors · 2025-11-23
>
> > **Q2**: Do the authors expect qualitatively different dynamics for different RLVR variants (e.g., GRPO, PPO, or CISPO), or would the theoretical conclusions remain consistent across these variants?
>
> **R2**: This is an excellent question! Yes, **while we used vanilla GRPO as our training algorithm for RLVR and RLIF in our experiments, our theoretical analysis can provide insights applicable to various RL algorithms, including PPO and GRPO**. Here's a detailed explanation of our theoretical framework and its connection to these RL variants:
> 1. Our theoretical analysis focuses on the optimization objective in Eq. (3.1):
>
>     $$\phi_{\mathrm{RL}}(\theta) = \mathbb{E}\_{x \sim \mathcal{D}, y \sim \pi\_{\theta}(\cdot \mid x)} \left[ r\_\phi(x, y) \right] - \beta \mathbb{D}\_{\mathrm{KL}}  \left[ \pi\_{\theta}(y \mid x)  \parallel \pi\_{\mathrm{ref}}(y \mid x) \right]$$
>    We consider the optimal case, i.e., the rollout number $N \rightarrow \infty$ such that we can directly optimize the expectation for $\phi_{\mathrm{RL}}(\theta)$ through policy gradient [1,2,4].
>     $$\frac{d}{d t}{\theta}(t) = \nabla \phi_{\mathrm{RL}} ({\theta} (t))$$
>     Many existing RL algorithms try to optimize $\phi_{\mathrm{RL}}(\theta)$ with limited data [3], resulting in various PPO-based algorithms including GRPO and CISPO, which introduce clipping terms in $\nabla \phi_{\mathrm{RL}} ({\theta} (t))$ and use data recycling for efficient, stable training. While our analysis considers the idealized gradient flow, it captures the core objective that all these policy gradient methods aim to optimize. Therefore, our theoretical framework provides insights that are broadly applicable to these variants.
>
> 2. In Proposition 5.2, we derive the optimal policy for both RLVR and RLIF. This result, which describes the convergence point of the model under different training objectives, is independent of the specific optimization algorithm used. It will therefore remain the same for different RL algorithms like PPO or GRPO. Furthermore, Theorems 5.3 and 5.4 demonstrate that RLVR's effectiveness is highly dependent on the base model's initialization, as it can either train effectively or get stuck at suboptimal points. This theoretical finding explains widely observed practical phenomena like the success of RL for LLMs is heavily dependent on the quality of the initial base model [5,6], and can provide further insights into efficient RL training, such as incorporating supervised finetuning before RL as a cold start [7]. Moreover, our theoretical analysis of RLIF provides an explanation for the initial performance gain of RLIF [8,9], and explains why the model's performance may decrease with continual training, which aligns with observations in [9].
>
>
>
> In conclusion, while our empirical results are based on GRPO, our theoretical analysis captures the fundamental principles of the RL objective, making it naturally extendable and insightful for various RLVR variants.
>
>
>
>
>
> [3] Schulman, John, et al. "Trust region policy optimization." _International conference on machine learning_. PMLR, 2015.
>
>
>
> [4] Sutton, Richard S., et al. "Policy gradient methods for reinforcement learning with function approximation." _Advances in neural information processing systems_ 12 (1999).
>
>
>
> [5] Zeng, Weihao, et al. "Simplerl-zoo: Investigating and taming zero reinforcement learning for open base models in the wild." _arXiv preprint arXiv:2503.18892_ (2025).
>
>
>
> [6] Gandhi, Kanishk, et al. "Cognitive behaviors that enable self-improving reasoners, or, four habits of highly effective stars." _arXiv preprint arXiv:2503.01307_ (2025).
>
>
>
> [7] Guo, Daya, et al. "Deepseek-r1: Incentivizing reasoning capability in llms via reinforcement learning." _arXiv preprint arXiv:2501.12948_ (2025).
>
>
>
> [8] Zhao, Xuandong, et al. "Learning to reason without external rewards." _arXiv preprint arXiv:2505.19590_ (2025).
>
>
>
> [9] Zhang, Yanzhi, et al. "No Free Lunch: Rethinking Internal Feedback for LLM Reasoning." _arXiv preprint arXiv:2506.17219_ (2025).

---

> ### Author Response · Authors · 2025-11-23
>
> >**Q3**Could the authors elaborate on Assumption 5.1?
>
> **R3**: Thank you for this insightful question! **While our assumption is primarily based on experimental observations, our theoretical framework can be extended to scenarios where reasoning pattern success rates change during training**. Let us address your specific questions:
>
> > **Q3.1**: How realistic is it to assume this remains constant
>
> **R3.1**:  This is an excellent point! **Our assumption is supported by empirical evidence from our experiments, and such stability is reasonable for current powerful large language models, and we have added additional experiments to further validate this observation**. Here we provide detailed explanation why such assumption is reasonable:
> 1. In our reasoning pattern level analysis, we used GPT-4 to extract and classify reasoning patterns in model responses. As shown in Fig. 1(b), the success rate for each reasoning pattern remains stable throughout the training process, while the pattern distribution changes during training. Moreover, we provide additional experiments to validate our observation, we kindly refer you to the last block of our response for detailed experimental results on different models and datasets. Recent works such as [10,11] also show that while the model's pass@K performance remains almost unchanged during training, RL will let the model choose CoT with higher success rates for better performance. Our findings align with these results.
> 2. This stability is particularly reasonable for current powerful large language models. When a specific reasoning pattern is chosen, these strong base models are unlikely to make basic errors (such as arithmetic mistakes). In such cases, improving the success rate of an individual reasoning pattern becomes challenging, as the model is already operating near its inherent capacity limit. Consequently, the optimization process naturally focuses on selecting more effective reasoning patterns rather than improving individual pattern success rates, which aligns with our observed training dynamics.
>
> > **Q3.2**: how might deviations affect the theoretical results?
>
> **R3.2**: This is a good question! The main role for Assumption 5.1 is theoretical analysis is too simplify our theoretical analysis, and provide a clear bound for training dynamics, **our theoretical framework and analysis can be extended to scenarios where reasoning pattern success rates changes during training**, here we explain how our theoretical analysis could extend to two different cases:
>
> - **case 1**. _Reasoning pattern success rates changes but the order for different reasoning patterns holds_, use $p^\*\_{\theta(t)}(r)$ denote the successrate for different reasoning patterns at step $t$, let $p^\*\_{\theta(t)}(r\_1) > p^\*\_{\theta(t)}(r\_2) > \cdots p^*\_{\theta(t)}(r\_n)$ holds for $t>0$, in this case, the condition for regim 1 in theorem 5.7 will holds across the training procedure, based on Eq D.5 the probabiity model choose the opeimal reasoning pattern $r_1$ will increase steality to reach $1$, resulting effective training, similarly, the condition for regim 2 in theorem 5.8 will holds until to regim 1, in this case, the probability for model to choose the suboptimal pattern remains to a relative high probability based on Eq D.6, resulting long time required to transform from regime 2 to regime 1.
>
> - **case 2**. reasoning pattern success rates change to reach their inherent capacity limit. Initially, RL would focus on improving the success rate of individual patterns. We can analyze this phase using the techniques from Regime 1 in Theorem 5.3. For a given reasoning pattern with a current success probability $p(t)$ and a maximum potential success rate $p^\*$, the optimization dynamics would first drive $p(t)$ towards $p^\*$. Our analysis shows the time to reach $|p^\* - p(t)| \leq \epsilon$ is $\mathcal{O}(1/\epsilon)$. Once the success rates of the primary reasoning patterns approach their respective limits (i.e., $\epsilon$ is sufficiently small), the main avenue for further reward improvement becomes shifting the probability of selecting *better* patterns. At this point, the training dynamic transitions into the scenario described by Assumption 5.1, and our main analysis holds for this crucial, later stage of training.
>
> Therefore, even if the assumption is relaxed, our theoretical framework remains relevant. It describes the key optimization phase that occurs after the model has learned to execute individual reasoning patterns proficiently, and our theoretical analysis can provide insights to more complicated rl training dynamics.
>
> [10] Yue, Yang, et al. Does reinforcement learning really incentivize reasoning capacity in llms beyond the base model?. arXiv:2504.13837
>
> [11] Wen, Xumeng, et al. Reinforcement learning with verifiable rewards implicitly incentivizes correct reasoning in base llms. arXiv:2506.14245

---

> ### Author Response · Authors · 2025-11-23
>
> > **Q4:** How sensitive are the observed "sparse token optimization" effects to the choice of base model or training dataset?
>
> **R4** This is a good question! To address your concern, we conducted additional experiments and observed "sparse token optimization" across different base models (Qwen, LLaMA) and training datasets (MATH, GSM8K, CodeContest). We report the ranking shift evaluated on the MATH dataset with sampling temperature = 0.4:
>
> | Method           | Step20 | Step40 | Step60 | Step80 | Step100 |
> |------------------|--------|--------|--------|--------|---------|
> | gsm8k-qwen-rlif  | 0.055  | 0.059  | 0.062  | 0.066  | 0.069   |
> | code-qwen-rlif   | 0.059  | 0.062  | 0.066  | 0.068  | 0.070   |
> | gsm8k-llama-rlif | 0.066  | 0.069  | 0.070  | 0.070  | 0.071   |
> | math-llama-rlvr  | 0.067  | 0.070  | 0.071  | 0.071  | 0.073   |
> | math-llama-rlif  | 0.062  | 0.066  | 0.067  | 0.067  | 0.067   |
> | gsm8k-llama-rlvr | 0.086  | 0.086  | 0.084  | 0.084  | 0.084   |
> | gsm8k-qwen-rlvr  | 0.056  | 0.057  | 0.057  | 0.058  | 0.057   |
>
>  where, "gsm8k-qwen-rlif" refers to Qwen-2.5-3B trained on GSM8K using RLIF. "gsm8k-llama-rlif" refers to Llama3.2-3b-Instruct trained on GSM8K using RLIF.
>
> We observe that with less than 10% of tokens experiencing a rank shift in most cases across different base models and training datasets, the experiments demonstrate that "sparse token optimization" is an inherent characteristic of RL training. RL optimizes only a small subset of tokens, reshaping the model's reasoning pattern selection process rather than changing the success rate of individual reasoning patterns. This provides experimental validation for our theoretical framework and analysis.

---

> ### Author Response · Authors · 2025-11-23
>
> **Additional Experiments: Reasoning Pattern Accuracy Across RLVR/RLIF Trained on Different Base Models and Datasets**
>
> We conducted RLIF and RLVR training across different base models (Qwen, LLaMA) and training datasets (MATH, GSM8K, CodeContest). We sampled questions from the MATH dataset to check the corresponding pattern accuracy. Since different base models exhibit different reasoning patterns, we selected reasoning patterns with the top-3 frequencies and report their accuracy throughout training.
>
> **LLama3.2-3B-Instruct as base model**:
> >
> > _**MATH-RLIF**_
> >
> >| pattern-acc    | base   | step20 | step40 | step60 | step80 | step100 |
> >|--------------|--------|--------|--------|--------|--------|---------|
> >| pattern1 acc | 0.505  | 0.497  | 0.514  | 0.499  | 0.506  | 0.505   |
> >| pattern2 acc | 0.094  | 0.085  | 0.085  | 0.066  | 0.085  | 0.057   |
> >| pattern3 acc | 0.080  | 0.089  | 0.103  | 0.103  | 0.085  | 0.094   |
> >
> > _**MATH-RLVR**_
> >
> >| patterns-acc     | base   | step20 | step40 | step60 | step80 | step100 |
> >|--------------|--------|--------|--------|--------|--------|---------|
> >| pattern1 acc | 0.505  | 0.497  | 0.514  | 0.499  | 0.506  | 0.505   |
> >| pattern2 acc | 0.094  | 0.085  | 0.085  | 0.085  | 0.094  | 0.094   |
> >| pattern3 acc | 0.080  | 0.113  | 0.089  | 0.080  | 0.141  | 0.117   |
> >
> >_**GSM8K-RLIF**_
> >
> >| patterns     | base   | step20 | step40 | step60 | step80 | step100 |
> >|--------------|--------|--------|--------|--------|--------|---------|
> >| pattern1 acc | 0.505  | 0.5135 | 0.539  | 0.5135 | 0.5177 | 0.495   |
> >| pattern2 acc | 0.094  | 0.085  | 0.123  | 0.047  | 0.076  | 0.085   |
> >| pattern3 acc | 0.080  | 0.080  | 0.127  | 0.094  | 0.084  | 0.136   |
> >
> >_**GSM8K-RLVR**_
> >
> >| patterns     | base   | step20 | step40 | step60 | step80 | step100     |
> >|--------------|--------|--------|--------|--------|--------|-------------|
> >| pattern1 acc | 0.505  | 0.461  | 0.484 | 0.526 | 0.516 | 0.511      |
> >| pattern2 acc | 0.094 | 0.1033 | 0.093 | 0.12   | 0.1333 | 0.113      |
> >| pattern3 acc | 0.080  | 0.094  | 0.070  | 0.080  | 0.075  | 0.103 |
>
> **Qwen2.5-3B as base model**:
> >
> > _**GSM8K-RLIF**_
> >
> > | patterns     | base   | step20 | step40 | step60 | step80 | step100 |
> > |--------------|--------|--------|--------|--------|--------|---------|
> > | pattern1 acc | 0.3355 | 0.3617 | 0.3377 | 0.366  | 0.3725 | 0.39    |
> > | pattern2 acc | 0.294  | 0.329  | 0.321  | 0.334  | 0.334  | 0.358   |
> > | pattern3 acc | 0.2021 | 0.2074 | 0.1915 | 0.234  | 0.234  | 0.2128  |
> >
> > _**GSM8K-RLVR**_
> >
> > | patterns     | base   | step20 | step40 | step60 | step80 | step100 |
> > |--------------|--------|--------|--------|--------|--------|---------|
> > | pattern1 acc | 0.3355 | 0.3529 | 0.3747 | 0.3834 | 0.3834 | 0.4139  |
> > | pattern2 acc | 0.294  | 0.332  | 0.350  | 0.385  | 0.347  | 0.369   |
> > | pattern3 acc | 0.2021 | 0.234  | 0.2713 | 0.2074 | 0.2181 | 0.2074  |
> >
> > _**Codecontest-RLIF**_, For this experiment, we use 50 code tasks sampled from HumanEval for evaluation, including training process performance analysis and reasoning pattern analysis.
> >
> > | patterns     | base   | step20 | step40 | step60 | step80 | step100 |
> > |--------------|--------|--------|--------|--------|--------|---------|
> > | pattern1 acc | 0.398  | 0.393  | 0.401  | 0.401  | 0.401  | 0.390   |
> > | pattern2 acc | 0.108  | 0.099  | 0.099  | 0.162  | 0.342  | 0.180   |
> > | pattern3 acc | 0.756  | 0.779  | 0.814  | 0.826  | 0.727  | 0.721   |
>
> We observe that reasoning pattern accuracy remains stable during training, further validating our experimental observations and supporting Assumption 5.1.

---

> ### Comment · Reviewer_xtTG · 2025-11-27
> **Response to Rebuttal**
>
> I thank the authors for addressing my questions and for running additional experiments. They provided evidence that Assumption 5.1 holds and that their analysis is applicable to different/newer RLVR variants. The authors also added important details and hyperparameters for their experiments, which improves reproducibility. I will therefore increase my score.

---

### Meta-Review · Area_Chair_MKeF · 2026-01-07

**Summary:**

This paper analyzes a surprising phenomenon in reinforcement learning-based fine-tuning of language models: that changes are primarily located in a sparse subset of "critical" token locations. Theoretical and empirical analysis combine to help deepen our understanding of how this process works.

The main reviewer complaints of the initial submission were related to the limited scope of the experimentation. I think this has been resolved by the expanded experiments done during the rebuttal, which show similar patterns with other base models and benchmark tasks. I think both the reviewers, in the discussion which remains, and I are satisfied by the rebuttal process, and I am happy to recommend acceptance for this paper.

----

It is worth noting that this submission is closely related to the separate, parallel submission https://openreview.net/forum?id=8vWIXno8LW. In the final version, you should cite this paper and discuss the differences.

**Reviewer Concerns:**

All seem resolved to me.

**Reviewer Scores:**

Scores were updated, and some increased significantly, during the discussion.

---

### Decision · Program_Chairs · 2026-01-26

Accept (Poster)